# Demystifying and Generalizing BinaryConnect

**Tim Dockhorn** *
University of Waterloo

**Yaoliang Yu**
University of Waterloo

**Eyyüb Sari**
Huawei Noah's Ark Lab

**Mahdi Zolnouri**
Huawei Noah's Ark Lab

**Vahid Partovi Nia**
Huawei Noah's Ark Lab

## Abstract

BinaryConnect (BC) and its many variations have become the de facto standard for neural network quantization. However, our understanding of the inner workings of BC is still quite limited. We attempt to close this gap in four different aspects: (a) we show that existing quantization algorithms, including post-training quantization, are surprisingly similar to each other; (b) we argue for proximal maps as a natural family of quantizers that is both easy to design and analyze; (c) we refine the observation that BC is a special case of dual averaging, which itself is a special case of the generalized conditional gradient algorithm; (d) consequently, we propose *ProxConnect* (PC) as a generalization of BC and we prove its convergence properties by exploiting the established connections. We conduct experiments on CIFAR-10 and ImageNet, and verify that PC achieves competitive performance.

## 1 Introduction

Scaling up to extremely large datasets and models has been a main ingredient for the success of deep learning. Indeed, with the availability of big data, more computing power, convenient software, and a bag of training tricks as well as algorithmic innovations, the size of models that we routinely train in order to achieve state-of-the-art performance has exploded, e.g., to billions of parameters in recent language models [9]. However, high memory usage and computational cost at inference time has made it difficult to deploy these models in real-time or on resource-limited devices [28]. The environmental impact of training and deploying these large models has also been recognized [40]. A common approach to tackle these problems is to compress a large model through quantization, i.e., replacing high-precision parameters with lower-precision ones. For example, we may constrain a subset of the weights to be binary [1, 5, 11, 29, 33, 37, 45] or ternary [26, 53]. Quantization can drastically decrease the carbon footprint of training and inference of neural networks, however, it may come at the cost of increased bias [21].

One of the main methods to obtain quantized neural networks is to encourage quantized parameters during gradient training using explicit or implicit regularization techniques, however, other methods are possible [16–18, 20, 25, 32, 51]. Besides the memory benefits, the structure of the quantization can speed up inference using, for example, faster matrix-vector products [18, 23]. Training and inference can be made even more efficient by also quantizing the activations [37] or gradients [52]. Impressive performance has been achieved with quantized networks, for example, on object detection [44] and natural language processing [43] tasks. The theoretical underpinnings of quantized neural networks, such as when and why their performance remains reasonably well, have been actively studied [3, 13, 22, 41, 46].

---

*Work done during an internship at Huawei Noah's Ark Lab. Correspondence to tim.dockhorn@uwaterloo.ca.

35th Conference on Neural Information Processing Systems (NeurIPS 2021).

BinaryConnect [BC, 11] and its many variations [10, 37, 53] are considered the gold standard for neural network quantization. Compared to plain (stochastic) gradient descent, BC does not evaluate the gradient at the current iterate but rather at a (close-by) quantization point using the Straight Through Estimator [6]. Despite its empirical success, BC has largely remained a "training trick" [1] and a rigorous understanding of its inner workings has yet to be found, with some preliminary steps taken in Li et al. [27] for the convex setting and in Yin et al. [45] for a particular quantization set. As pointed out in Bai et al. [5], BC only evaluates gradients at the finite set of quantization points, and therefore does not exploit the rich information carried in the continuous weights network. Bai et al. [5] also observed that BC is formally equivalent to the dual averaging algorithm [31, 42], while some similarity to the mirror descent algorithm was found in Ajanthan et al. [1].

The main goal of this work is to significantly improve our understanding of BC, by connecting it with well-established theory and algorithms. In doing so we not only simplify and improve existing results but also obtain novel generalizations. We summarize our main contributions in more details:

- In Section 2, we show that existing gradient-based quantization algorithms are surprisingly similar to each other: the only high-level difference is at what points we evaluate the gradient and perform the update.

- In Section 3, we present a principled theory for constructing proximal quantizers. Our results unify previous efforts, remove tedious calculations, and bring theoretical convenience. We illustrate our theory by effortlessly designing a new quantizer that can be computed in one-pass, works for different quantization sets (binary or multi-bit), and includes previous attempts as special cases [5, 11, 45, 53].

- In Section 4, we significantly extend the observation of Bai et al. [5] that the updates of BC are the same as the dual averaging algorithm [31, 42]: BC is a nonconvex counterpart of dual averaging, and more importantly, dual averaging itself is simply the generalized conditional gradient algorithm applied to a smoothened dual problem. The latter fact, even in the convex case, does not appear to be widely recognized to the best of our knowledge.

- In Section 5, making use of the above established results, we propose *ProxConnect* (PC) as a family of algorithms that generalizes BC and we prove its convergence properties for both the convex and the nonconvex setting. We rigorously justify the diverging parameter in proximal quantizers and resolve a discrepancy between theory and practice in the literature [1, 5, 45].

- In Section 6, we verify that PC outperforms BC and ProxQuant [5] on CIFAR-10 for both fine-tuning pretrained models as well as end-to-end training. On the more challenging ImageNet dataset, PC yields competitive performance despite of minimal hyperparameter tuning.

## 2 Background

We consider the usual (expected) objective $\ell(\mathbf{w}) = \mathbf{E}\ell(\mathbf{w}, \boldsymbol{\xi})$, where $\boldsymbol{\xi}$ represents random sampling. For instance, $\ell(\mathbf{w}) = \frac{1}{n}\sum_{i=1}^{n}\ell_i(\mathbf{w})$ where $\boldsymbol{\xi}$ is uniform over $n$ training samples (or minibatches thereof), $\mathbf{w}$ are the weights of a neural network and $\ell_i$ may be the cross-entropy loss (of the $i$-th training sample). We denote a sample (sub)gradient of $\ell$ at $\mathbf{w}$ and $\boldsymbol{\xi}$ as $\widetilde{\nabla}\ell(\mathbf{w}) = \nabla\ell(\mathbf{w}, \boldsymbol{\xi})$ so that $\mathbf{E}\widetilde{\nabla}\ell(\mathbf{w}) = \nabla\ell(\mathbf{w})$. Throughout, we use the starred notation $\mathbf{w}^*$ for continuous weights and reserve $\mathbf{w}$ for (semi)discrete ones.

We are interested in solving the following (nonconvex) problem:

$$\min_{\mathbf{w} \in Q} \ell(\mathbf{w}), \tag{1}$$

where $Q \subseteq \mathbb{R}^d$ is a discrete, nonconvex quantization set. For instance, on certain low-resource devices it may be useful or even mandatory to employ binary weights, i.e., $Q = \{\pm 1\}^d$. Importantly, our goal is to compete against the non-quantized, continuous weights network (i.e. $Q = \mathbb{R}^d$). In other words, we do *not* necessarily aim to solve (the hard, combinatorial) problem (1) globally and optimally. Instead, we want to find discrete weights $\mathbf{w} \in Q$ that remain satisfactory when compared to the non-quantized continuous weights. This is how we circumvent the difficulty in (1) and more importantly how the structure of $\ell$ could come into aid. If $\ell$ is reasonably smooth, a close-by quantized weight of a locally, or even globally, optimal continuous weight will likely yield similar performance.

Tremendous progress has been made on quantizing and compressing neural networks. While it is not possible to discuss all related work, below we recall a few families of gradient-based quantization algorithms that directly motivate our work; more methods can be found in recent surveys [15, 36].

**BinaryConnect (BC).** Courbariaux et al. [11] considered binary networks where $Q = \{\pm 1\}^d$ and proposed the BinaryConnect algorithm:

$$\mathbf{w}_{t+1}^* = \mathbf{w}_t^* - \eta_t \widetilde{\nabla}\ell(\mathsf{P}(\mathbf{w}_t^*)), \tag{2}$$

where $\mathsf{P}$ is a projector that quantizes the continuous weights $\mathbf{w}_t^*$ either deterministically (by taking the sign) or stochastically. Note that the (sample) gradient is evaluated at the quantized weights $\mathbf{w}_t := \mathsf{P}(\mathbf{w}_t^*)$, while its continuous output, after scaled by the step size $\eta_t$, is added to the continuous weights $\mathbf{w}_t^*$. For later comparison, it is useful to break the BC update into the following two pieces:

$$\mathbf{w}_t = \mathsf{P}(\mathbf{w}_t^*), \qquad \mathbf{w}_{t+1}^* = \mathbf{w}_t^* - \eta_t \widetilde{\nabla}\ell(\mathbf{w}_t). \tag{3}$$

Other choices of $\mathsf{P}$ [1, 45] and $Q$ [44, 53] in this framework have also been experimented with.

**ProxQuant (PQ).** Bai et al. [5] applied the usual proximal gradient to solve (1):

$$\mathbf{w}_{t+1} = \mathsf{P}(\mathbf{w}_t - \eta_t \widetilde{\nabla}\ell(\mathbf{w}_t)), \tag{4}$$

which can be similarly decomposed into:

$$\mathbf{w}_t = \mathsf{P}(\mathbf{w}_t^*), \qquad \mathbf{w}_{t+1}^* = \mathbf{w}_t - \eta_t \widetilde{\nabla}\ell(\mathbf{w}_t). \tag{5}$$

Thus, the only high-level difference between BC and PQ is that the former updates the continuous weights $\mathbf{w}_t^*$ while the latter updates the quantized weights $\mathbf{w}_t$. This seemingly minor difference turns out to cause drastically different behaviors of the two algorithms. For example, choosing $\mathsf{P}$ to be the Euclidean projection to $Q$ works well for BinaryConnect but not at all for ProxQuant.

**Reversing BinaryConnect.** The above comparison naturally suggests a reversed variant of BC:

$$\mathbf{w}_t = \mathsf{P}(\mathbf{w}_t^*), \qquad \mathbf{w}_{t+1}^* = \mathbf{w}_t - \eta_t \widetilde{\nabla}\ell(\mathbf{w}_t^*), \tag{6}$$

which amounts to switching the continuous and quantized weights in the BC update. Similar to PQ, the choice of $\mathsf{P}$ is critical in this setup. To the best of our knowledge, this variant has not been formally studied before. Reversing BinaryConnect may not be intuitive, however, it serves as a starting point for a more general method (see Section 5).

**Post-Training Quantization.** Lastly, we can also rewrite the naive post-training quantization scheme in a similar form:

$$\mathbf{w}_t = \mathsf{P}(\mathbf{w}_t^*), \qquad \mathbf{w}_{t+1}^* = \mathbf{w}_t^* - \eta_t \widetilde{\nabla}\ell(\mathbf{w}_t^*), \tag{7}$$

where we simply train the continuous network as usual and then quantize at the end. Note that the quantized weights $\mathbf{w}_t$ do not affect the update of the continuous weights $\mathbf{w}_t^*$.

# 3  What Makes a Good Quantizer?

As we have seen in Section 2, the choice of the quantizer $\mathsf{P}$ turns out to be a crucial element for solving (1). Indeed, if $\mathsf{P} = \mathsf{P}_Q$ is the projector onto the discrete quantization set $Q$, then BC (and ProxQuant) only evaluate the gradient of $\ell$ at (the finite set of) points in $Q$. As a result, the methods will not be able to exploit the rich information carried in the continuous weights network, which can lead to non-convergence [Fig. 1b, 5]. Since then, many semi-discrete quantizers, that turn continuous weights into more and more discrete ones, have been proposed [1, 5, 33, 45]. In this section, we present a principled way to construct quantizers that unifies previous efforts, removes tedious calculations, and also brings theoretical convenience when it comes down to analyzing the convergence of the algorithms in Section 2.

Our construction is based on the proximal map $\mathsf{P}_r^\mu : \mathbb{R}^d \rightrightarrows \mathbb{R}^d$ of a (closed) function $r$:

$$\mathsf{P}_r^\mu(\mathbf{w}^*) = \underset{\mathbf{w}}{\arg\min} \ \frac{1}{2\mu}\|\mathbf{w} - \mathbf{w}^*\|_2^2 + r(\mathbf{w}), \tag{8}$$

where $\mu > 0$ is a smoothing parameter. The proximal map is well-defined as long as the function $r$ is lower bounded by a quadratic function, in particular, when $r$ is bounded from below. If $r$ is proper and (closed) convex, then the minimizer on the right-hand side of (8) is uniquely attained, while for general (nonconvex) functions the proximal map may be multi-valued (hence the notation $\rightrightarrows$). If $r = \iota_Q$ is the indicator function (see (15) below), then $\mathsf{P}_r^\mu$ reduces to the familiar projector $\mathsf{P}_Q$ (for any $\mu$). Remarkably, a complete characterization of such maps on the real line is available:

**Theorem 3.1** ([Proposition 3, 48]). A (possibly multi-valued) map $\mathsf{P}\colon \mathbb{R} \rightrightarrows \mathbb{R}$ is a proximal map (of some function r) iff it is (nonempty) compact-valued, monotone and has a closed graph. The underlying function r is unique (up to addition of constants) iff $\mathsf{P}$ is convex-valued, while r is convex iff $\mathsf{P}$ is nonexpansive (i.e. 1-Lipschitz continuous).

The sufficient and necessary conditions of Theorem 3.1 allow one to design proximal maps $\mathsf{P}$ directly, without needing to know the underlying function r at all (even though it is possible to integrate a version of r from $\mathsf{P}$). The point is that, as far as quantization algorithms are concerned, having $\mathsf{P}$ is enough, and hence one is excused from the tedious calculations in deriving $\mathsf{P}$ from r as is typical in existing works.

For example, the (univariate) mirror maps constructed in Ajanthan et al. [Theorem 1, 1], such as $\tanh$, are proximal maps according to Theorem 3.1: In our notation, the (Mirror Descent) updates of Ajanthan et al. [1] are the same as (3) with $\mathsf{P} = (\nabla \Phi)^{-1}$ for some mirror map $\Phi$. Since $\Phi$ is taken to be strictly convex in Ajanthan et al. [1], it is easy to verify that $\mathsf{P}$ satisfies all conditions of Theorem 3.1.[2]

The next result allows us to employ *stochastic* quantizers, where in each iteration we randomly choose one of $\mathsf{P}_i$ to quantize the weights[3]. We may also apply different quantizers to different layers of a neural network.

**Theorem 3.2.** Let $\mathsf{P}_i : \mathbb{R}^d \rightrightarrows \mathbb{R}^d, i = [k]$, be proximal maps. Then, the averaged map

$$\mathsf{P} := \sum_{i=1}^{k} \alpha_i \mathsf{P}_i, \qquad \text{where } \alpha_i \geq 0, \;\; \sum_{i=1}^{k} \alpha_i = 1, \tag{9}$$

is also a proximal map. Similarly, the product map

$$\mathsf{P} := \mathsf{P}_1 \times \mathsf{P}_2 \times \cdots \times \mathsf{P}_k, \;\; \mathbf{w}^* = (\mathbf{w}_1^*, \ldots, \mathbf{w}_k^*) \mapsto \big(\mathsf{P}_1(\mathbf{w}_1^*), \ldots, \mathsf{P}_k(\mathbf{w}_k^*)\big) \tag{10}$$

is a proximal map (from $\mathbb{R}^{dk}$ to $\mathbb{R}^{dk}$).

(The proof of Theorem 3.2 and all other omitted proofs can be found in Appendix A.)

**Example 3.3.** Let $Q$ be a quantization set (e.g. $Q = \{-1, 0, 1\}$). Clearly, the identity map id and the projector $\mathsf{P}_Q$ are proximal maps. Therefore, their convex combination $\mathsf{P}^\mu = \frac{\mathsf{id} + \mu \mathsf{P}_Q}{1+\mu}, \mu \geq 0$, is also a proximal map, which is exactly the quantizer used in Yin et al. [45].

Lastly, we mention that it is intuitively desirable to have $\mathsf{P}^\mu \to \mathsf{P}_Q$ when $\mu$ increases indefinitely, so that the quantizer eventually settles on *bona fide* discrete values in $Q$. This is easily achieved by letting minimizers of the underlying function r, or the (possibly larger set of) fixed points of $\mathsf{P}^\mu$, approach $Q$. We give one such construction by exploiting Theorem 3.1 and Theorem 3.2.

### 3.1 General Piecewise Linear Quantizers

Let $Q_j = \{q_{j,k}\}_{k=1}^{b_j}$ (with $q_{j,1} \leq \cdots \leq q_{j,b_j}$) be the quantization set for the $j$-th weight group[4] $\mathbf{w}_j^* \in \mathbb{R}^{d_j}$. Let $p_{j,k+1} := \frac{q_{j,k} + q_{j,k+1}}{2}, k \in [b_j - 1]$, be the middle points. We introduce two parameters $\rho, \varrho \geq 0$ and define

$$\text{horizontal shifts:} \qquad q_{j,k}^- := p_{j,k} \vee (q_{j,k} - \rho), \qquad q_{j,k}^+ := p_{j,k+1} \wedge (q_{j,k} + \rho), \tag{11}$$

$$\text{vertical shifts:} \qquad p_{j,k+1}^- := q_{j,k} \vee (p_{j,k+1} - \varrho), \;\; p_{j,k+1}^+ := q_{j,k+1} \wedge (p_{j,k+1} + \varrho), \tag{12}$$

with $q_{j,1}^- = q_{j,1}$ and $q_{j,b_j}^+ = q_{j,b_j}$. Then, we define $\mathsf{L}_\rho^\varrho$ as the piece-wise linear map (that simply connects the points by straight lines):

$$\mathsf{L}_\rho^\varrho(w^*) := \begin{cases} q_{j,k}, & \text{if } q_{j,k}^- \leq w^* \leq q_{j,k}^+ \\ q_{j,k} + (w^* - q_{j,k}^+)\frac{p_{j,k+1}^- - q_{j,k}}{p_{j,k+1} - q_{j,k}^+}, & \text{if } q_{j,k}^+ \leq w^* < p_{j,k+1} \\ p_{j,k+1}^+ + (w^* - p_{j,k+1})\frac{q_{j,k+1} - p_{j,k+1}^+}{q_{j,k+1}^- - p_{j,k+1}}, & \text{if } p_{j,k+1} < w^* \leq q_{j,k+1}^-, \end{cases} \tag{13}$$

---

[2]The above reasoning hinges on $\Phi$ being univariate. More generally, if a multivariate mirror map $\Phi$ is 1-strongly convex (as is typical in Mirror Descent), then it follows from Moreau [Corollaire 10.c, 30] that $\mathsf{P} = (\nabla \Phi)^{-1}$, being a nonexpansion, is again a proximal map (of some convex function).

[3]This form of stochasticity still leads to *determinisitc networks*, and is therefore conceptually different from *probabilistic* (quantized) networks [35, 39].

[4]In this section, time subscripts are not needed. Instead, we use subscripts to indicate weight groups.

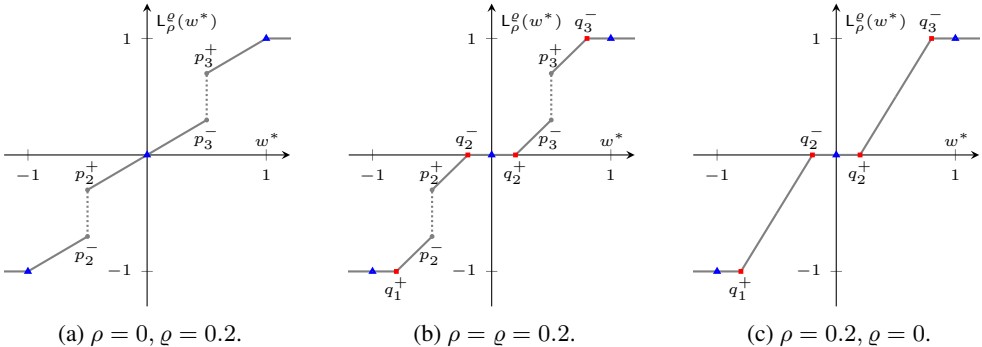

|  (a) $\rho = 0$, $\varrho = 0.2$. | (b) $\rho = \varrho = 0.2$. | (c) $\rho = 0.2$, $\varrho = 0$. |

Figure 1: Different instantiations of the proximal map $\mathsf{L}_\rho^\varrho$ in (13) for $Q = \{-1, 0, 1\}$.

for all $w^* \in \mathbf{w}_j^*$. At the middle points, $\mathsf{L}_\rho^\varrho(p_{j,k+1})$ may take any value within the two limits. Following the commonly used weight clipping in BC [11], we may set $\mathsf{L}_\rho^\varrho(w^*) = q_{j,1}$ for $w^* < q_{j,1}$ and $\mathsf{L}_\rho^\varrho(w^*) = q_{j,b_j}$ for $w^* > q_{j,b_j}$, however, other choices may also work well. The proximal map $\mathsf{L}_\rho^\varrho$ of an example ternary component is visualized in Figure 1 for different choices of $\rho$ and $\varrho$.

The (horizontal) parameter $\rho$ controls the discretization vicinity within which a continuous weight will be pulled *exactly* into the discrete set $Q_j$, while the (vertical) parameter $\varrho$ controls the slope (i.e. expansiveness) of each piece. It follows at once from Theorem 3.1 that $\mathsf{L}_\rho^\varrho$ is indeed a proximal map. In particular, setting $\rho = 0$ (hence continuous weights are only discretized in the limit) and $\varrho = \frac{\mu}{2(1+\mu)}$ we recover Example 3.3 (assuming w.l.o.g. that $q_{j,k+1} - q_{j,k} \equiv 1$). On the other hand, setting $\rho = \varrho$ leads to a generalization of the (binary) quantizer in Bai et al. [5], which keeps the slope to the constant 1 (while introducing jumps at middle points) and happens to be the proximal map of the distance function to $Q_j$ [5]. Of course, setting $\rho = \varrho = 0$ yields the identity map and allows us to skip quantizing certain weights (as is common practice), while letting $\rho, \varrho \to \infty$ recovers the projector $\mathsf{P}_{Q_j}$.

Needless to say, we may adapt the quantization set $Q_j$ and the parameters $\varrho$ and $\rho$ for different weight groups, creating a multitude of quantization schemes. By Theorem 3.2, the overall operator remains a proximal map.

## 4 Demystifying BinaryConnect (BC)

Bai et al. [5] observed that the updates of BC are formally the same as the dual averaging (DA) algorithm [31, 42], even though the latter algorithm was originally proposed and analyzed only for convex problems. A lesser known fact is that (regularized) dual averaging itself is a special case of the generalized conditional gradient (GCG) algorithm. In this section, we first present the aforementioned fact, refine the observation of Bai et al. [5], and set up the stage for generalizing BC.

### 4.1 Generalized Conditional Gradient is Primal-Dual

We first present the generalized conditional gradient (GCG) algorithm [8, 49] and point out its ability to solve simultaneously the primal and dual problems.

Let us consider the following "regularized" problem:

$$\min_{\mathbf{w} \in \mathbb{R}^d} \; f(\mathbf{w}) := \ell(\mathbf{w}) + \mathsf{r}(\mathbf{w}), \tag{14}$$

where $\mathsf{r}$ is a general (nonconvex) regularizer. Setting $\mathsf{r}$ to the indicator function of $Q$, i.e.,

$$\mathsf{r}(\mathbf{w}) = \iota_Q(\mathbf{w}) = \begin{cases} 0, & \text{if } \mathbf{w} \in Q \\ \infty, & \text{otherwise} \end{cases}, \tag{15}$$

reduces (14) to the original problem (1). As we will see, incorporating an arbitrary $\mathsf{r}$ does not add any complication but will allow us to immediately generalize BC.

Introducing the Fenchel conjugate function $\ell^*$ (resp. $\mathsf{r}^*$) of $\ell$ (resp. $\mathsf{r}$):

$$\ell^*(\mathbf{w}^*) := \sup_{\mathbf{w}} \; \langle \mathbf{w}, \mathbf{w}^* \rangle - \ell(\mathbf{w}), \tag{16}$$

which is always (closed) convex even when $\ell$ itself is nonconvex, we state the Fenchel–Rockafellar dual problem [38]:

$$\min_{\mathbf{w}^* \in \mathbb{R}^d} \ell^*(-\mathbf{w}^*) + \mathsf{r}^*(\mathbf{w}^*), \tag{17}$$

which, unlike the original problem (14), is always a convex problem.

We apply the generalized conditional gradient algorithm [8, 49] to solving the dual problem[5] (17): Given $\mathbf{w}_t^*$, we linearize the function $\mathsf{r}^*$ and solve

$$\mathbf{z}_t^* = \left[ \arg\min_{\mathbf{w}^*} \ell^*(-\mathbf{w}^*) + \langle \mathbf{w}^*, \mathbf{w}_t \rangle \right] = -\nabla\ell^{**}(\mathbf{w}_t), \quad \mathbf{w}_t := \nabla\mathsf{r}^*(\mathbf{w}_t^*), \tag{18}$$

where we used the fact that $(\nabla\ell^*)^{-1} = \nabla\ell^{**}$. Then, we take the convex combination

$$\mathbf{w}_{t+1}^* = (1 - \lambda_t)\mathbf{w}_t^* + \lambda_t\mathbf{z}_t^*, \quad \text{where } \lambda_t \in [0, 1]. \tag{19}$$

The following theorem extends Bach [Proposition 4.2, 4] and Yu [Theorem 4.6, 47] to any $\lambda_t$:

**Theorem 4.1.** Suppose $\mathsf{r}^*$ is $L$-smooth (i.e. $\nabla\mathsf{r}^*$ is $L$-Lipschitz continuous), then for any $\mathbf{w}$:

$$\sum_{\tau=0}^{t} \frac{\lambda_\tau}{\pi_\tau} [(\ell^{**} + \mathsf{r}^{**})(\mathbf{w}_\tau) - (\ell^{**} + \mathsf{r}^{**})(\mathbf{w})] \le (1 - \lambda_0)\Delta(\mathbf{w}, \mathbf{w}_0) + \sum_{\tau=0}^{t} \frac{\lambda_\tau^2}{2\pi_\tau} L\|\mathbf{w}_\tau^* - \mathbf{z}_\tau^*\|_2^2, \tag{20}$$

where $\mathbf{w}_t := \nabla\mathsf{r}^*(\mathbf{w}_t^*)$, $\pi_t := \prod_{\tau=1}^{t}(1 - \lambda_\tau)$, and $\pi_0 := 1$. $\Delta(\mathbf{w}, \mathbf{w}_t) := \mathsf{r}^{**}(\mathbf{w}) - \mathsf{r}^{**}(\mathbf{w}_t) - \langle \mathbf{w} - \mathbf{w}_t, \mathbf{w}_t^* \rangle$ is the Bregman divergence induced by the convex function $\mathsf{r}^{**}$.

While the convergence of $\mathbf{w}_t^*$ to the minimum of (17) is well-known (see e.g. Yu et al. [49]), the above result also implies that a properly averaged iterate $\bar{\mathbf{w}}_t$ also converges to the minimum of the dual problem of (17):

**Corollary 4.2.** Let $\bar{\mathbf{w}}_t := \sum_{\tau=0}^{t} \Lambda_{t,\tau}\mathbf{w}_\tau$, where $\Lambda_{t,\tau} := \frac{\lambda_\tau}{\pi_\tau}/H_t$ and $H_t := \sum_{\tau=0}^{t} \frac{\lambda_\tau}{\pi_\tau}$. Then, we have for any $\mathbf{w}$:

$$(\ell^{**} + \mathsf{r}^{**})(\bar{\mathbf{w}}_t) - (\ell^{**} + \mathsf{r}^{**})(\mathbf{w}) \le \frac{(1 - \lambda_0)\Delta(\mathbf{w}, \mathbf{w}_0)}{H_t} + \frac{L}{2}\sum_{\tau=0}^{t} \lambda_\tau\Lambda_{t,\tau}\|\mathbf{w}_\tau^* - \mathbf{z}_\tau^*\|_2^2. \tag{21}$$

Assuming $\{\mathbf{z}_\tau^*\}$ is bounded (e.g. when $\ell^{**}$ is Lipschitz continuous), the right-hand side of (21) diminishes if $\lambda_t \to 0$ and $\sum_t \lambda_t = \infty$. Setting $\lambda_t = \frac{1}{t+1}$ recovers ergodic averaging $\bar{\mathbf{w}}_t = \frac{1}{t+1}\sum_{\tau=0}^{t} \mathbf{w}_\tau$ for which the right-hand side of (21) diminishes at the rate[6] $O(\log t/t)$; see Appendix A.2 for details.

Thus, GCG solves problem (17) and its dual simultaneously.

## 4.2 BC $\subseteq$ DA $\subseteq$ GCG

We are now in a position to reveal the relationships among BinaryConnect (BC), (regularized) dual averaging (DA) and the generalized conditional gradient (GCG). Since Theorem 4.1 requires $\mathsf{r}^*$ to be $L$-smooth, in the event that it is not we resort to a smooth approximation known as the Moreau envelope [30]:

$$\mathcal{M}_{\mathsf{r}^*}^\mu(\mathbf{w}^*) = \min_{\mathbf{z}^*} \frac{1}{2\mu}\|\mathbf{w}^* - \mathbf{z}^*\|_2^2 + \mathsf{r}^*(\mathbf{z}^*), \tag{22}$$

where the minimizer is (by definition) exactly $\mathsf{P}_{\mathsf{r}^*}^\mu(\mathbf{w}^*)$. It is well-known that $\mathcal{M}_{\mathsf{r}^*}^\mu$ is $(1/\mu)$-smooth and $(\mathcal{M}_{\mathsf{r}^*}^\mu)^* = \mathsf{r}^{**} + \frac{\mu}{2}\|\cdot\|_2^2$ [30]. We then apply GCG to the approximate dual problem:

$$\min_{\mathbf{w}^* \in \mathbb{R}^d} \ell^*(-\mathbf{w}^*) + \mathcal{M}_{\mathsf{r}^*}^\mu(\mathbf{w}^*), \tag{23}$$

whose own Fenchel–Rockfellar dual is:

$$\left[ \min_{\mathbf{w} \in \mathbb{R}^d} \ell^{**}(\mathbf{w}) + (\mathcal{M}_{\mathsf{r}^*}^\mu)^*(\mathbf{w}) \right] = \min_{\mathbf{w} \in \mathbb{R}^d} \ell^{**}(\mathbf{w}) + \mathsf{r}^{**}(\mathbf{w}) + \frac{\mu}{2}\|\mathbf{w}\|_2^2. \tag{24}$$

---

[5]GCG is usually applied to solving the primal problem (14) directly. Our choice of the dual problem here is to facilitate later comparison with dual averaging and BinaryConnect.

[6]The log factor can be removed by setting, for example, $\lambda_t = \frac{2}{2+t}$ instead.

The updates of GCG applied to the approximate problem (23) are thus:

$$\mathbf{w}_t \coloneqq \nabla \mathcal{M}_{r^*}^\mu(\mathbf{w}_t^*) = \mathsf{P}_{r^{**}}^{1/\mu}(\mathbf{w}_t^*/\mu) \quad \text{(see Proposition A.2 for derivation)} \tag{25}$$

$$\mathbf{w}_{t+1}^* = (1 - \lambda_t)\mathbf{w}_t^* + \lambda_t \mathbf{z}_t^*, \quad \text{where} \quad \mathbf{z}_t^* = -\nabla \ell^{**}(\mathbf{w}_t), \tag{26}$$

which is exactly the updates of (regularized) dual averaging [42] for convex problems where[7] $r^{**} = r$ and $\ell^{**} = \ell$. Nesterov [31] motivated dual averaging by the natural desire of non-decreasing step sizes, whereas conventional subgradient algorithms "counter-intuitively" assign smaller step sizes to more recent iterates instead. Based on our explanation, we conclude this is possible because dual averaging solves an (approximate) smoothened dual problem, hence we can afford to use a constant (rather than a diminishing/decreasing) step size.

Defining $\pi_t \coloneqq \prod_{s=1}^t (1 - \lambda_s)$ for $t \geq 1$, $\pi_0 \coloneqq 1$, $\pi_{-1} \coloneqq (1 - \lambda_0)^{-1}$, and setting $\mathsf{w}_t^* = \mathbf{w}_t^*/\pi_{t-1}$, we have:

$$\mathbf{w}_t = \mathsf{P}_{r^{**}}^{1/\mu}(\pi_{t-1}\mathsf{w}_t^*/\mu), \qquad \mathsf{w}_{t+1}^* = \mathsf{w}_t^* - \tfrac{\lambda_t}{\pi_t}\nabla \ell^{**}(\mathbf{w}_t). \tag{27}$$

Let us now reparameterize

$$\eta_t \coloneqq \tfrac{\lambda_t}{\pi_t} \implies \lambda_t = \tfrac{\eta_t}{1+\sum_{\tau=1}^t \eta_\tau} \text{ and } \tfrac{1}{\pi_t} = 1 + \sum_{\tau=1}^t \eta_\tau, \tag{28}$$

for $t \geq 1$. If we also allow $\mu = \mu_t$ to change adaptively from iteration to iteration (as in dual averaging), in particular, if $\mu_t = \pi_{t-1}$, we obtain the familiar update:

$$\mathbf{w}_t = \mathsf{P}_{r^{**}}^{1/\pi_{t-1}}(\mathsf{w}_t^*), \qquad \mathsf{w}_{t+1}^* = \mathsf{w}_t^* - \eta_t \nabla \ell^{**}(\mathbf{w}_t). \tag{29}$$

For nonconvex problems, we may replace $r^{**}$ and $\ell^{**}$ with their nonconvex counterparts $r$ and $\ell$, respectively. We remark that $r^{**}$ (resp. $\ell^{**}$) is the largest convex function that is (pointwise) majorized by $r$ (resp. $\ell$). In particular, with $r = \iota_Q$ and $\mathsf{P}_r^{1/\mu} = \mathsf{P}_Q$ (for any $\mu$) we recover the BinaryConnect update (3). While the parameter $\mu$ plays no role when $r$ is an indicator function, we emphasize that for general $r$ we should use the quantizer $\mathsf{P}_r^{1/\pi_{t-1}}$, where importantly $1/\pi_{t-1} \to \infty$ hence the quantizer converges to minimizers of $r$ asymptotically. Neglecting this crucial detail may lead to suboptimality as is demonstrated in the following example:

**Example 4.3.** Bai et al. [5] constructed the following intriguing example:

$$\ell(w) = \tfrac{1}{2}w^2, \ \ Q = \{\pm 1\}, \ \ \mathsf{P}_r^{1/\mu}(w) = \text{sign}(w)\frac{\epsilon|w|+\frac{1}{\mu}}{\epsilon+\frac{1}{\mu}} \text{ for } |w| \leq 1, \tag{30}$$

and they showed non-convergence of the algorithm $w \leftarrow w - \eta\nabla\ell(\mathsf{P}_r^{1/\mu}(w))$, where $\mu$ is a fixed constant. If we use $\mathsf{P}_r^{1/\pi_{t-1}}$ with some diverging $1/\pi_{t-1}$ instead, the resulting BinaryConnect, with diminishing $\eta_t$ or ergodic averaging, would actually converge to 0 (since $\mathsf{P}_r^{1/\pi_{t-1}} \to \text{sign}$).

## 5 ProxConnect (PC): A Generalization of BinaryConnect

We are now ready to generalize BC by combining the results from Section 3 and Section 4. Replacing the convex envelopes, $\ell^{**}$ and $r^{**}$, with their nonconvex counterparts and replacing deterministic gradients with stochastic gradients (as well as the change-of-variable $\mathsf{w}_t^* \to \mathbf{w}_t^*$), we obtain from (29) a *family* of algorithms which we term ProxConnect (PC):

$$\mathbf{w}_t = \mathsf{P}_r^{1/\pi_{t-1}}(\mathbf{w}_t^*), \quad \mathbf{w}_{t+1}^* = \mathbf{w}_t^* - \eta_t \widetilde{\nabla}\ell(\mathbf{w}_t), \tag{31}$$

where the quantizer $\mathsf{P}_r^{1/\pi_{t-1}}$ may be designed directly by following Section 3. We have already seen in Section 4 that BC belongs to PC by choosing $\mathsf{P}_r^{1/\pi_{t-1}} = \mathsf{P}_Q$ (in which case $\pi_{t-1}$ plays no role).

The analysis in Section 4, initially tailored to convex functions, immediately generalizes to the nonconvex algorithm PC (for nonconvex $\ell$, nonconvex $r$, and stochastic gradients $\widetilde{\nabla}\ell$):

---

[7]That is, if we set $\lambda_t = 1/t$ and allow for time-dependent $\mu_t = t$; see Xiao [Algorithm 1, 42].

**Theorem 5.1.** Fix any $\mathbf{w}$, the iterates in (31) satisfy:

$$\sum_{\tau=s}^{t} \eta_\tau[\langle \mathbf{w}_\tau - \mathbf{w}, \widetilde{\nabla}\ell(\mathbf{w}_\tau)\rangle + \mathsf{r}(\mathbf{w}_\tau) - \mathsf{r}(\mathbf{w})] \le \Delta_{s-1}(\mathbf{w}) - \Delta_t(\mathbf{w}) + \sum_{\tau=s}^{t} \Delta_\tau(\mathbf{w}_\tau), \qquad (32)$$

where $\Delta_\tau(\mathbf{w}) := \mathsf{r}_\tau(\mathbf{w}) - \mathsf{r}_\tau(\mathbf{w}_{\tau+1}) - \langle \mathbf{w} - \mathbf{w}_{\tau+1}, \mathbf{w}_{\tau+1}^* \rangle$ is the Bregman divergence induced by the (possibly nonconvex) function $\mathsf{r}_\tau(\mathbf{w}) := \frac{1}{\pi_\tau}\mathsf{r}(\mathbf{w}) + \frac{1}{2}\|\mathbf{w}\|_2^2$.

The summand on the left-hand side of (32) is related to the duality gap in Yu et al. [49], which is a natural measure of stationarity for the nonconvex problem (14). Indeed, it reduces to the familiar ones when convexity is present:

**Corollary 5.2.** For convex $\ell$ and any $\mathbf{w}$, the iterates in (31) satisfy:

$$\min_{\tau=s,\dots,t} \mathbf{E}[f(\mathbf{w}_\tau) - f(\mathbf{w})] \le \tfrac{1}{\sum_{\tau=s}^{t}\eta_\tau} \cdot \mathbf{E}\big[\Delta_{s-1}(\mathbf{w}) - \Delta_t(\mathbf{w}) + \sum_{\tau=s}^{t} \Delta_\tau(\mathbf{w}_\tau)\big]. \qquad (33)$$

If $\mathsf{r}$ is also convex, then

$$\min_{\tau=s,\dots,t} \mathbf{E}[f(\mathbf{w}_\tau) - f(\mathbf{w})] \le \tfrac{1}{\sum_{\tau=s}^{t}\eta_\tau} \cdot \mathbf{E}\big[\Delta_{s-1}(\mathbf{w}) + \sum_{\tau=s}^{t} \tfrac{\eta_\tau^2}{2}\|\widetilde{\nabla}\ell(\mathbf{w}_\tau)\|_2^2\big], \qquad (34)$$

and

$$\mathbf{E}\big[f(\bar{\mathbf{w}}_t) - f(\mathbf{w})\big] \le \tfrac{1}{\sum_{\tau=s}^{t}\eta_\tau} \cdot \mathbf{E}\big[\Delta_{s-1}(\mathbf{w}) + \sum_{\tau=s}^{t} \tfrac{\eta_\tau^2}{2}\|\widetilde{\nabla}\ell(\mathbf{w}_\tau)\|_2^2\big], \qquad (35)$$

where $\mathbf{w}_t = \frac{\sum_{\tau=s}^{t}\eta_\tau \mathbf{w}_\tau}{\sum_{\tau=s}^{t}\eta_\tau}$.

The right-hand sides of (34) and (35) diminish iff $\eta_t \to 0$ and $\sum_t \eta_t = \infty$ (assuming boundedness of the stochastic gradient). We note some trade-off in choosing the step size $\eta_\tau$: both the numerator and denominator of the right-hand sides of (34) and (35) are increasing w.r.t. $\eta_\tau$. The same conclusion can be drawn for (33) and (32), where $\Delta_\tau$ also depends on $\eta_\tau$ (through the accumulated magnitude of $\mathbf{w}_{\tau+1}^*$). A detailed analysis may need to take specific properties of $\mathsf{r}$ or $\mathsf{P}$ into account [45].

**ProxQuant vs ProxConnect.** It is worthwhile to point out one important difference between Prox-Quant and ProxConnect: Bai et al. [5] proved convergence (to some notion of stationarity) of ProxQuant for a fixed quantizer (see Bai et al. [Theorem 5.1, 5]), i.e., $\mathsf{P}_\mathsf{r}^\mu$ for a fixed $\mu$, but their experiments relied on incrementing $\mu$ so that their quantizer approaches the projector $\mathsf{P}_Q$. This creates some discrepancy between theory and practice. The same comment also applies to [1]. In contrast, ProxConnect is derived from a rigorous theory that automatically justifies a diverging $\mu$. In particular, choosing a constant step size $\eta_\tau \equiv \eta_0$ would lead to $1/\pi_{t-1} \propto t$, matching the current practice that is now justifiable if $\mathsf{r}$ is strongly convex; see Appendix A.3.1 for details.

**Connection to Existing Algorithms.** Besides the obvious BC, ProxConnect generalizes many other quantization algorithms. As it turns out, many of these algorithms can also be realized using our proposed proximal quantizer $\mathsf{L}_\rho^\varrho$ from Section 3; see Table 1 for a sample summary.

**reverseProxConnect.** In Section 2, we discussed the idea of reversing BinaryConnect. As Prox-Connect generalizes BC, we also present reverseProxConnect (rPC) as a generalization of reversing BinaryConnect:

$$\mathbf{w}_t = \mathsf{P}_\mathsf{r}^{1/\pi_{t-1}}(\mathbf{w}_t^*), \quad \mathbf{w}_{t+1}^* = \mathbf{w}_t - \eta_t \widetilde{\nabla}\ell(\mathbf{w}_t^*), \qquad (36)$$

In contrast to reversing BinnaryConnect, rPC is not completely without merit: it evaluates the gradient at the continuous weights $\mathbf{w}_t^*$ and hence is able to exploit a richer landscape of the loss. Even when stuck at a fixed discrete weight $\mathbf{w}_t$, rPC may still accumulate sizable updates (as long as the step size and the gradient remain sufficiently large) to allow it to eventually jump out of $\mathbf{w}_t$: note that the continuous weights $\mathbf{w}_t^*$ still get updated. Finally, for constant step size $\eta$ and $\pi$, we note that fixed points of rPC, when existent, satisfy:

$$\mathbf{w}^* = \mathsf{P}_\mathsf{r}^{1/\pi}(\mathbf{w}^*) - \eta\nabla\ell(\mathbf{w}^*) \iff \mathbf{w}^* = (\mathsf{id} + \eta\nabla\ell)^{-1}\left(\mathsf{id} + \eta\left[\tfrac{\partial\mathsf{r}}{\pi\eta}\right]\right)^{-1}(\mathbf{w}^*) \qquad (37)$$

$$=: \mathcal{B}\left(\eta, \nabla\ell, \tfrac{\partial\mathsf{r}}{\pi\eta}\right)(\mathbf{w}^*), \qquad (38)$$

Table 1: A sample summary of existing quantization algorithms. The PC column indicates if the method is a special case of our proposed ProxConnect algorithm. The $\mathsf{L}_\rho^\varrho$-column indicates if the method uses a quantizer which is a special case of our general quantizer $\mathsf{L}_\rho^\varrho$ introduced in Section 3. If so, the $\rho, \varrho$-column states how $\rho$ and $\varrho$ were chosen (in practice): increasing ↗, fixed to 0, or fixed to ∞. Other than ProxQuant-Ternary, all methods can compute their quantizers in a single neural network pass. †: TrainedTernary methods might use a quantizer different than $\mathsf{L}_\rho^\varrho$ to improve performance.

| Method | PC | One-pass | Learnable parameters | $\mathsf{L}_\rho^\varrho$ | $\rho, \varrho$ |
|---|---|---|---|---|---|
| ProxQuant-Binary-$L_1$ [5] | ✗ | ✓ | ✗ | ✓ | ↗, ↗ |
| ProxQuant-Ternary [5] | ✗ | ✗ | ✗ | ✗ | - |
| BinaryConnect [11] | ✓ | ✓ | ✗ | ✓ | ∞,∞ |
| BinaryRelax [45] | ✓ | ✓ | ✗ | ✓ | 0, ↗ |
| BinaryWeight [37] | ✓ | ✓ | ✓ | ✗ | - |
| MirrorDescentView [1] | ✓ | ✓ | ✗ | ✗ | - |
| TrainedTernary† [53] | ✓ | ✓ | ✓ | ✓ | ∞,∞ |
| TernaryWeight [26] | ✓ | ✓ | ✓ | ✗ | - |
| ProxConnect (ours) | - | ✓ | ✓ | ✓ | ↗, ↗ |

where for simplicity we assumed deterministic gradients $\nabla\ell$ and $\mathsf{r}$ to be convex such that $\mathsf{P}_\mathsf{r}^{1/\pi} = (\mathsf{id} + \partial\mathsf{r}/\pi)^{-1}$. The operator $\mathcal{B}$ is known as the backward-backward update (as opposed to the forward-backward update in ProxQuant), and it is known that when $\eta \to 0$ slowly, backward-backward updates converge to a stationary point [34]. Thus, despite of our current limited understanding of rPC, there is some reason (and empirical evidence as shown in Section 6) to believe it might still be interesting.

**Importance of GCG framework:** Deriving PC from the GCG framework may let us transfer recent advances on GCG [7, 14, 50] to neural network quantization. Furthermore, it is the cornerstone in justifying the widely-adopted diverging smoothing parameter.

## 6 Experiments

### 6.1 Classification on CIFAR-10

We perform image classification on CIFAR-10 [24] using ResNet20 and ResNet56 [19], comparing BinaryConnect [11] with ProxQuant [5] and (reverse)ProxConnect using our proposed proximal operator $\mathsf{L}_\rho^\varrho$. For fair comparison, we set $\rho = \varrho$ in $\mathsf{L}_\rho^\varrho$ as this resembles the quantizer (for binary quantization) used in the original ProxQuant algorithm. Similar to Bai et al. [5], we increase the parameter $\rho$ (or equivalently $\varrho$) linearly: $\rho_t = (1 + t/B)\rho_0$. In contrast to Bai et al. [5], however, we increase $\rho$ after every gradient step rather than after every epoch as this is more in line with our analysis. We treat $\rho_0$ as a hyperparameter for which we conduct a small grid search. We consider binary ($Q = \{-1, 1\}$), ternary ($Q = \{-1, 0, 1\}$) and quaternary ($Q = \{-1, -0.3, 0.3, 1\}$) quantization. Details for all CIFAR-10 experiments can be found in Appendix B.1.

**Fine-Tuning Pretrained Models.** In this experiment, all quantization algorithms are initialized with a pretrained ResNet. The test accuracies of the pretrained ResNet20 and ResNet56 are 92.01 and 93.01, respectively. Table 2 shows the final test accuracies for the different models. For ProxQuant and (reverse)ProxConnect we respectively picked the best $\rho_0$ values; results for all $\rho_0$ values can be found in Appendix B.1.7. (Reverse)ProxConnect outperforms the other two methods on all six settings.

**End-To-End Training.** To save computational costs, it is important that quantization algorithms also perform well when they are not initialized with pretrained full-precision model. We therefore compare the four methods for randomly initialized models; see Table 3 for the results. ProxConnect outperforms all other methods on all six tasks. Interestingly, ProxQuant and reverseProxConnect perform considerably worse for all six tasks when compared to fine-tuning. The performance drop of BinaryConnect and ProxConnect when compared to fine-tuning is only significant for ternary

Table 2: Fine-tuning pretrained ResNets. Final test accuracy: mean and standard deviation in 3 runs.

| Model | Quantization | BC [11] | PQ [5] | rPC (ours) | PC (ours) |
|---|---|---|---|---|---|
| ResNet20 | Binary | **90.31** (0.00) | 89.94 (0.10) | 89.98 (0.17) | **90.31** (0.21) |
| | Ternary | 74.95 (0.16) | 91.46 (0.06) | **91.47** (0.19) | 91.37 (0.18) |
| | Quaternary | 91.43 (0.07) | 91.43 (0.21) | 91.43 (0.06) | **91.81** (0.14) |
| ResNet56 | Binary | 92.22 (0.12) | 92.33 (0.06) | 92.47 (0.29) | **92.65** (0.16) |
| | Ternary | 74.68 (1.4) | 93.07 (0.02) | 92.84 (0.11) | **93.25** (0.12) |
| | Quaternary | 93.20 (0.06) | 92.82 (0.16) | 92.91 (0.26) | **93.42** (0.12) |

Table 3: End-to-end training of ResNets. Final test accuracy: mean and standard deviation in 3 runs.

| Model | Quantization | BC [11] | PQ [5] | rPC (ours) | PC (ours) |
|---|---|---|---|---|---|
| ResNet20 | Binary | 87.51 (0.21) | 81.59 (0.75) | 81.82 (0.32) | **89.92** (0.65) |
| | Ternary | 27.10 (0.21) | 47.98 (1.30) | 47.17 (1.94) | **84.09** (0.16) |
| | Quaternary | 89.91 (0.09) | 85.29 (0.09) | 85.05 (0.27) | **90.17** (0.14) |
| ResNet56 | Binary | 89.79 (0.45) | 86.13 (1.71) | 86.25 (1.50) | **91.26** (0.59) |
| | Ternary | 30.31 (7.79) | 50.54 (3.68) | 42.95 (1.57) | **84.36** (0.75) |
| | Quaternary | 90.69 (0.57) | 87.81 (1.60) | 87.30 (1.02) | **91.70** (0.14) |

quantization. We found that ProxQuant and reverseProxConnect can be quite sensitive to the choice of $\rho_0$, whereas ProxConnect is stable in this regard; see Appendix B.1.7.

## 6.2 Classification on ImageNet

We perform a small study on ImageNet [12] using ResNet18 [19]. As can be seen in Table 4, BC performs slightly better for fine-tuning whereas PC performs slightly better for end-to-end training. This is not an exhaustive study, but rather a first indication that PC can yield competitive performance on large scale datasets. For more details on the experiments see Appendix B.2.

Table 4: Fine-tuning (left) and end-to-end training (right). Final test accuracy: mean and standard deviation over three runs.

| BC [11] | PC (ours) | | BC [11] | PC (ours) | |
|---|---|---|---|---|---|
| | $\rho_0 = 2e{-}2$ | $\rho_0 = 4e{-}2$ | | $\rho_0 = 2.5e{-}3$ | $\rho_0 = 5e{-}3$ |
| **65.84** (0.04) | 65.44 (0.13) | 65.70 (0.04) | 63.79 (0.12) | **63.89** (0.14) | 63.67 (0.12) |

## 7 Conclusion

Capitalizing on a principled approach for designing quantizers and a surprising connection between BinaryConnect and the generalized conditional gradient (GCG) algorithm, we proposed ProxConnect as a unification and generalization of existing neural network quantization algorithms. Our analysis refines prior convergence guarantees and our experiments confirm the competitiveness of ProxConnect. In future work, we plan to apply ProxConnect to training other models such as transformers. The connection with GCG also opens the possibility for further acceleration.

## Acknowledgments and Disclosure of Funding

We thank the anonymous reviewers for their constructive comments as well as the area chair and the senior area chair for overseeing the review process. YY thanks NSERC for funding support. We thank Jingjing Wang for some early discussions.

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
