# A Proofs

In this section we include all proofs omitted in the main paper, and supply some additional comments.

## A.1 Proofs for Section 3

**Theorem 3.2.** Let $\mathsf{P}_i : \mathbb{R}^d \rightrightarrows \mathbb{R}^d, i = [k]$, be proximal maps. Then, the averaged map

$$\mathsf{P} := \sum_{i=1}^{k} \alpha_i \mathsf{P}_i, \qquad \text{where } \alpha_i \geq 0, \quad \sum_{i=1}^{k} \alpha_i = 1, \tag{9}$$

is also a proximal map. Similarly, the product map

$$\mathsf{P} := \mathsf{P}_1 \times \mathsf{P}_2 \times \cdots \times \mathsf{P}_k, \quad \mathbf{w}^* = (\mathbf{w}_1^*, \ldots, \mathbf{w}_k^*) \mapsto \big(\mathsf{P}_1(\mathbf{w}_1^*), \ldots, \mathsf{P}_k(\mathbf{w}_k^*)\big) \tag{10}$$

is a proximal map (from $\mathbb{R}^{dk}$ to $\mathbb{R}^{dk}$).

*Proof.* Define $\mathsf{P}(\mathbf{w}) = \sum_i \alpha_i \mathsf{P}_i(\mathbf{w})$ if each $\mathsf{P}_i(\mathbf{w})$ is single-valued. It is known that any proximal map $\mathsf{P}_i$ is almost everywhere single-valued [38], thus $\mathsf{P}$ is almost everywhere defined. Then, if necessary we take the closure of the graph of $\mathsf{P}$ so that it is defined everywhere. (This last step can be omitted if we always take the upper or lower limits at any jump of $\mathsf{P}_i$.) With this interpretation of the average in (9), the rest of the first claim then follows from [Proposition 4, 48].

For the second claim about the product map in (10), let $\mathsf{P}_i$ be the proximal map of $f_i(\mathbf{w}_i^*)$. Then, it follows immediately from the definition (8) that the product map $\mathsf{P}$ is the proximal map of the sum function $f(\mathbf{w}^*) := \sum_i f_i(\mathbf{w}_i^*)$, where $\mathbf{w}^* = (\mathbf{w}_1^*, \ldots, \mathbf{w}_k^*)$. $\qquad\square$

## A.2 Proofs for Section 4

Let us first record a general result for iterates of the following type:

$$\mathsf{w}_{t+1}^* = \mathsf{w}_t^* + \eta_t \mathbf{z}_t^*, \tag{39}$$

where $\mathsf{w}_0^*$ is given and $\mathbf{z}_t^*$ may be arbitrary.

**Lemma A.1.** For any $\mathbf{w}$, any sequence of $\mathbf{w}_t$, and arbitrary function $g$, the iterates (39) satisfy:

$$\sum_{\tau=s}^{t} \eta_\tau [\langle \mathbf{w}_\tau - \mathbf{w}, -\mathbf{z}_\tau^* \rangle + g(\mathbf{w}_\tau) - g(\mathbf{w})] = \delta_{s-1}(\mathbf{w}) - \delta_t(\mathbf{w}) + \sum_{\tau=s}^{t} \delta_\tau(\mathbf{w}_\tau), \tag{40}$$

where $\delta_\tau(\mathbf{w}) := \frac{1}{\pi_\tau} g(\mathbf{w}) - \frac{1}{\pi_\tau} g(\mathbf{w}_{\tau+1}) - \langle \mathbf{w} - \mathbf{w}_{\tau+1}, \mathsf{w}_{\tau+1}^* \rangle$ and $\frac{1}{\pi_t} = 1 + \sum_{\tau=1}^{t} \eta_\tau$.

*Proof.* The proof is simple algebra (and was discovered by abstracting the original, tedious proof of Theorem 4.1). For any $\mathbf{w}$, we verify:

$$\sum_{\tau=s}^{t} \eta_\tau [\langle \mathbf{w}_\tau - \mathbf{w}, -\mathbf{z}_\tau^* \rangle + g(\mathbf{w}_\tau) - g(\mathbf{w})] \tag{41}$$

$$= \sum_{\tau=s}^{t} \langle \mathbf{w}_\tau - \mathbf{w}, \mathsf{w}_\tau^* - \mathsf{w}_{\tau+1}^* \rangle + \eta_\tau [g(\mathbf{w}_\tau) - g(\mathbf{w})] \tag{42}$$

$$= \sum_{\tau=s}^{t} \langle \mathbf{w}_\tau - \mathbf{w}, \mathsf{w}_\tau^* - \mathsf{w}_{\tau+1}^* \rangle + (\tfrac{1}{\pi_\tau} - \tfrac{1}{\pi_{\tau-1}})[g(\mathbf{w}_\tau) - g(\mathbf{w})] \tag{43}$$

$$= \sum_{\tau=s}^{t} - \langle \mathbf{w}_\tau - \mathbf{w}_{\tau+1}, \mathsf{w}_{\tau+1}^* \rangle + \langle \mathbf{w} - \mathbf{w}_{\tau+1}, \mathsf{w}_{\tau+1}^* \rangle - \langle \mathbf{w} - \mathbf{w}_\tau, \mathsf{w}_\tau^* \rangle + (\tfrac{1}{\pi_\tau} - \tfrac{1}{\pi_{\tau-1}})[g(\mathbf{w}_\tau) - g(\mathbf{w})] \tag{44}$$

$$= \sum_{\tau=s}^{t} \delta_\tau(\mathbf{w}_\tau) - \delta_\tau(\mathbf{w}) + \delta_{\tau-1}(\mathbf{w}). \tag{45}$$

Telescoping completes the proof. $\qquad\square$

**Theorem 4.1.** Suppose $r^*$ is $L$-smooth (i.e. $\nabla r^*$ is $L$-Lipschitz continuous), then for any $\mathbf{w}$:

$$\sum_{\tau=0}^{t} \tfrac{\lambda_\tau}{\pi_\tau}[(\ell^{**} + r^{**})(\mathbf{w}_\tau) - (\ell^{**} + r^{**})(\mathbf{w})] \le (1 - \lambda_0)\Delta(\mathbf{w}, \mathbf{w}_0) + \sum_{\tau=0}^{t} \tfrac{\lambda_\tau^2}{2\pi_\tau} L\|\mathbf{w}_\tau^* - \mathbf{z}_\tau^*\|_2^2, \quad (20)$$

where $\mathbf{w}_t := \nabla r^*(\mathbf{w}_t^*)$, $\pi_t := \prod_{\tau=1}^{t}(1 - \lambda_\tau)$, and $\pi_0 := 1$. $\Delta(\mathbf{w}, \mathbf{w}_t) := r^{**}(\mathbf{w}) - r^{**}(\mathbf{w}_t) - \langle \mathbf{w} - \mathbf{w}_t, \mathbf{w}_t^* \rangle$ is the Bregman divergence induced by the convex function $r^{**}$.

*Proof.* We first expand the recursion

$$\mathbf{w}_{t+1}^* = (1 - \lambda_t)\mathbf{w}_t^* + \lambda_t \mathbf{z}_t^* \tag{46}$$

$$= (1 - \lambda_t)(1 - \lambda_{t-1})\mathbf{w}_{t-1}^* + \lambda_t \mathbf{z}_t^* + (1 - \lambda_t)\lambda_{t-1}\mathbf{z}_{t-1}^* \tag{47}$$

$$= \dots \tag{48}$$

$$= (1 - \lambda_0)\pi_t \mathbf{w}_0^* + \sum_{\tau=0}^{t} \tfrac{\pi_t}{\pi_\tau}\lambda_\tau \mathbf{z}_\tau^*, \tag{49}$$

where $\pi_t := \prod_{\tau=1}^{t}(1 - \lambda_\tau)$ for $t \ge 1$ while $\pi_0 := 1$ and $\lambda_t \in [0, 1]$. We deduce (by, for instance, setting $\mathbf{z}_t^* = \mathbf{w}_t^* \equiv 1$, for all $t \ge 0$, in (49)) that

$$\frac{1}{\pi_t} = (1 - \lambda_0) + \sum_{\tau=0}^{t} \frac{\lambda_\tau}{\pi_\tau} = 1 + \sum_{\tau=1}^{t} \frac{\lambda_\tau}{\pi_\tau}. \tag{50}$$

Define $\frac{1}{\pi_{-1}} := 1 - \lambda_0$ and $\mathbf{z}_t^* = -\nabla \ell^{**}(\mathbf{w}_t)$. We apply the convexity of $\ell^{**}$ to the LHS of (20):

$$\text{LHS} = \sum_{\tau=0}^{t} \tfrac{\lambda_\tau}{\pi_\tau}[\ell^{**}(\mathbf{w}_\tau) + r^{**}(\mathbf{w}_\tau) - \ell^{**}(\mathbf{w}) - r^{**}(\mathbf{w})] \tag{51}$$

$$\le \sum_{\tau=0}^{t} \tfrac{\lambda_\tau}{\pi_\tau}[\langle \mathbf{w}_\tau - \mathbf{w}, -\mathbf{z}_\tau^* \rangle + r^{**}(\mathbf{w}_\tau) - r^{**}(\mathbf{w})]. \tag{52}$$

Next, we identify $\eta_\tau := \frac{\lambda_\tau}{\pi_\tau}$, $g = r^{**}$, $\mathbf{w}_{\tau+1}^* := \mathbf{w}_{\tau+1}^*/\pi_\tau = \mathbf{w}_\tau^* + \eta_\tau \mathbf{z}_\tau^*$, and $\delta_\tau = \frac{1}{\pi_\tau}\Delta$ so that we can apply Lemma A.1:

$$\text{LHS} \le \sum_{\tau=0}^{t} \tfrac{\lambda_\tau}{\pi_\tau}[\langle \mathbf{w}_\tau - \mathbf{w}, -\mathbf{z}_\tau^* \rangle + r^{**}(\mathbf{w}_\tau) - r^{**}(\mathbf{w})] \tag{53}$$

$$\le (1 - \lambda_0)\Delta(\mathbf{w}, \mathbf{w}_0) - \tfrac{1}{\pi_t}\Delta(\mathbf{w}, \mathbf{w}_{t+1}) + \sum_{\tau=0}^{t} \tfrac{1}{\pi_\tau}\Delta(\mathbf{w}_\tau, \mathbf{w}_{\tau+1}) \tag{54}$$

$$\le (1 - \lambda_0)\Delta(\mathbf{w}, \mathbf{w}_0) + \sum_{\tau=0}^{t} \tfrac{1}{\pi_\tau}\tfrac{L}{2}\|\mathbf{w}_{\tau+1}^* - \mathbf{w}_\tau^*\|_2^2, \tag{55}$$

where in the last step we applied the nonnegativity of the Bregman divergence $\Delta$ (when induced by a convex function such as $r^{**}$) as well as the following inequality:

$$\Delta(\mathbf{w}_\tau, \mathbf{w}_{\tau+1}) = r^{**}(\mathbf{w}_\tau) - r^{**}(\mathbf{w}_{\tau+1}) - \langle \mathbf{w}_\tau - \mathbf{w}_{\tau+1}, \mathbf{w}_{\tau+1}^* \rangle \tag{56}$$

$$= r^{**}(\mathbf{w}_\tau) - r^{**}(\mathbf{w}_{\tau+1}) - \langle \mathbf{w}_\tau - \mathbf{w}_{\tau+1}, \nabla r^{**}(\mathbf{w}_{\tau+1}) \rangle \quad (\mathbf{w}_t = \nabla r^*(\mathbf{w}_t^*) \iff \mathbf{w}_t^* = \nabla r^{**}(\mathbf{w}_t)) \tag{57}$$

$$= r^{***}(\mathbf{w}_{\tau+1}^*) - r^{***}(\mathbf{w}_\tau^*) - \langle \mathbf{w}_{\tau+1}^* - \mathbf{w}_\tau^*, \nabla r^{***}(\nabla r^{**}(\mathbf{w}_\tau)) \rangle \quad \text{(by duality of Bregman divergence)} \tag{58}$$

$$= r^*(\mathbf{w}_{\tau+1}^*) - r^*(\mathbf{w}_\tau^*) - \langle \mathbf{w}_{\tau+1}^* - \mathbf{w}_\tau^*, \nabla r^*(\nabla r^{**}(\mathbf{w}_\tau)) \rangle \quad \text{(by convexity of } r^{***}) \tag{59}$$

$$= r^*(\mathbf{w}_{\tau+1}^*) - r^*(\mathbf{w}_\tau^*) - \langle \mathbf{w}_{\tau+1}^* - \mathbf{w}_\tau^*, \mathbf{w}_\tau \rangle \quad \left(\text{since } \nabla r^{**} = (\nabla r^*)^{-1}\right) \tag{60}$$

$$\le \langle \mathbf{w}_{\tau+1}^* - \mathbf{w}_\tau^*, \nabla r^*(\mathbf{w}_\tau^*) \rangle + \tfrac{L}{2}\|\mathbf{w}_{\tau+1}^* - \mathbf{w}_\tau^*\|_2^2 - \langle \mathbf{w}_{\tau+1}^* - \mathbf{w}_\tau^*, \mathbf{w}_\tau \rangle \quad \text{(by smoothness of } r^*) \tag{61}$$

$$= \tfrac{L}{2}\|\mathbf{w}_{\tau+1}^* - \mathbf{w}_\tau^*\|_2^2 \quad \text{(since } \mathbf{w}_\tau := \nabla r^*(\mathbf{w}_\tau^*)). \tag{62}$$

Applying (46) completes the proof. $\qquad\square$

**Corollary 4.2.** Let $\bar{\mathbf{w}}_t := \sum_{\tau=0}^{t} \Lambda_{t,\tau} \mathbf{w}_\tau$, where $\Lambda_{t,\tau} := \frac{\lambda_\tau}{\pi_\tau}/H_t$ and $H_t := \sum_{\tau=0}^{t} \frac{\lambda_\tau}{\pi_\tau}$. Then, we have for any $\mathbf{w}$:

$$(\ell^{**} + r^{**})(\bar{\mathbf{w}}_t) - (\ell^{**} + r^{**})(\mathbf{w}) \leq \frac{(1-\lambda_0)\Delta(\mathbf{w}, \mathbf{w}_0)}{H_t} + \frac{L}{2} \sum_{\tau=0}^{t} \lambda_\tau \Lambda_{t,\tau} \|\mathbf{w}_\tau^* - \mathbf{z}_\tau^*\|_2^2. \quad (21)$$

*Proof.* By the convexity of $r^{**}$ and $\ell^{**}$ as well as the fact that the sum of two convex functions is convex, we have:

$$(\ell^{**} + r^{**})(\bar{\mathbf{w}}_t) \leq \sum_{\tau=0}^{t} \Lambda_{t,\tau}(\ell^{**} + r^{**})(\mathbf{w}_\tau). \quad (63)$$

Inserting the above in Theorem 4.1 (multiplied $1/H_t$) and noting that $\sum_{\tau=0}^{t} \Lambda_{t,\tau} = 1$, we have

$$(\ell^{**} + r^{**})(\bar{\mathbf{w}}_t) - (\ell^{**} + r^{**})(\mathbf{w}) \leq \sum_{\tau=0}^{t} \Lambda_{t,\tau} \left[ (\ell^{**} + r^{**})(\mathbf{w}_\tau) - (\ell^{**} + r^{**})(\mathbf{w}) \right] \quad (64)$$

$$\leq \frac{(1-\lambda_0)\Delta(\mathbf{w}, \mathbf{w}_0)}{H_t} + \frac{L}{2} \sum_{\tau=0}^{t} \lambda_\tau \Lambda_{t,\tau} \|\mathbf{w}_\tau^* - \mathbf{z}_\tau^*\|_2^2. \quad (65)$$

$\square$

The following instantiation of Corollary 4.2 is notable: setting $\lambda_t = \frac{1}{t+1}$ implies $\pi_t = \lambda_t$, $H_t = t+1$, and therefore $\bar{\mathbf{w}}_t$ is simply ergodic averaging, i.e.,

$$\bar{\mathbf{w}}_t = \frac{1}{t+1} \sum_{s=0}^{t} \mathbf{w}_t. \quad (66)$$

The right-hand side of (21) diminishes at the rate of $O(\frac{\log t}{t})$ since

$$\sum_{\tau=0}^{t} \lambda_t \Lambda_{t,\tau} = \frac{1}{t+1} \sum_{\tau=0}^{t} \frac{1}{\tau+1}, \quad \text{and} \quad \sum_{\tau=0}^{t} \frac{1}{\tau+1} = O(\log t). \quad (67)$$

We remark that the log factor can be removed if we set $\lambda_t = \frac{2}{t+2}$ instead, for which we have

$$\pi_t = \prod_{\tau=1}^{t} \frac{\tau}{\tau+2} = \frac{2}{(t+1)(t+2)}, \quad (68)$$

$$H_t = \sum_{\tau=0}^{t} \frac{\lambda_\tau}{\pi_\tau} = \sum_{\tau=0}^{t} \frac{2}{\tau+2} \frac{(\tau+1)(\tau+2)}{2} = \frac{1}{2}(t+1)(t+2), \quad (69)$$

$$\sum_{\tau=0}^{t} \lambda_t \Lambda_{t,\tau} = \frac{4}{(t+1)(t+2)} \sum_{\tau=0}^{t} \frac{\tau+1}{\tau+2}, \quad \text{and} \quad \sum_{\tau=0}^{t} \frac{\tau+1}{\tau+2} = \mathcal{O}(t). \quad (70)$$

**Proposition A.2.**

$$\nabla \mathscr{M}_{r^*}^{\mu}(\mathbf{w}_t^*) = \mathsf{P}_{r^{**}}^{1/\mu}(\mathbf{w}_t^*/\mu) \quad (71)$$

*Proof.* By the envelope theorem, we have $\nabla \mathscr{M}_{r^*}^{\mu}(\mathbf{w}_t^*) = \frac{\mathbf{w}_t^* - \mathsf{P}_{r^*}^{\mu}(\mathbf{w}_t^*)}{\mu}$. Furthermore, using the Moreau decomposition, we have

$$\mathbf{w}_t^* = \mathsf{P}_{\mu r^*}^{1}(\mathbf{w}_t^*) + \mathsf{P}_{(\mu r^*)^*}^{1}(\mathbf{w}_t^*) \quad (72)$$

$$= \mathsf{P}_{r^*}^{\mu}(\mathbf{w}_t^*) + \mathsf{P}_{\mathsf{s}}^{\mu}(\mathbf{w}_t^*), \quad (73)$$

where $s(\mathbf{w}) := r^{**}(\mathbf{w}/\mu)$. Combining the two results, we have

$$\nabla \mathscr{M}_{r^*}^\mu(\mathbf{w}_t^*) = \frac{\mathbf{w}_t^* - P_{r^*}^\mu(\mathbf{w}_t^*)}{\mu} \tag{74}$$

$$= \frac{P_s^\mu(\mathbf{w}_t^*)}{\mu} \tag{75}$$

$$= \tfrac{1}{\mu} \arg\min_{\mathbf{w}} \tfrac{1}{2\mu} \|\mathbf{w} - \mathbf{w}_t^*\|_2^2 + s(\mathbf{w}) \tag{76}$$

$$= \arg\min_{\mathbf{w}} \tfrac{1}{2\mu} \|\mu\mathbf{w} - \mathbf{w}_t^*\|_2^2 + s(\mu\mathbf{w}) \tag{77}$$

$$= \arg\min_{\mathbf{w}} \tfrac{\mu}{2} \|\mathbf{w} - \tfrac{\mathbf{w}_t^*}{\mu}\|_2^2 + r^{**}(\mathbf{w}) \tag{78}$$

$$= P_{r^{**}}^{1/\mu}(\mathbf{w}_t^*/\mu). \tag{79}$$

$\square$

### A.3 Proofs for Section 5

In fact, Theorem 5.1 is a special case of a more general result:

**Theorem A.3.** Given $\mathbf{w}_0^*$, $\eta_t \geq 0$ and $\frac{1}{\pi_t} = 1 + \sum_{\tau=1}^t \eta_\tau$, consider the iterates defined as

$$\mathbf{w}_t = P_r^{1/\mu_t}(\pi_{t-1}\mathbf{w}_t^*/\mu_t), \qquad \mathbf{w}_{t+1}^* = \mathbf{w}_t^* - \eta_t \widetilde{\nabla}\ell(\mathbf{w}_t). \tag{80}$$

Then, for any $\mathbf{w}$,

$$\sum_{\tau=s}^t \eta_\tau[\langle \mathbf{w}_\tau - \mathbf{w}, \widetilde{\nabla}\ell(\mathbf{w}_\tau)\rangle + r(\mathbf{w}_\tau) - r(\mathbf{w})] \leq \Delta_{s-1}(\mathbf{w}) - \Delta_t(\mathbf{w}) + \sum_{\tau=s}^t \Delta_\tau(\mathbf{w}_\tau) + \tag{81}$$

$$+ \sum_{\tau=s}^t \tfrac{1}{2}\big(\tfrac{\mu_{\tau+1}}{\pi_\tau} - \tfrac{\mu_\tau}{\pi_{\tau-1}}\big)(\|\mathbf{w}\|_2^2 - \|\mathbf{w}_\tau\|_2^2), \tag{82}$$

where $\Delta_\tau(\mathbf{w}) := r_\tau(\mathbf{w}) - r_\tau(\mathbf{w}_{\tau+1}) - \langle \mathbf{w} - \mathbf{w}_{\tau+1}, \mathbf{w}_{\tau+1}^* \rangle$ is the Bregman divergence induced by the (possibly nonconvex) function $r_\tau(\mathbf{w}) := \frac{1}{\pi_\tau}r(\mathbf{w}) + \frac{\mu_{\tau+1}}{2\pi_\tau}\|\mathbf{w}\|_2^2$.

*Proof.* We simply apply Lemma A.1 with $\mathbf{z}_\tau^* := -\widetilde{\nabla}\ell(\mathbf{w}_\tau)$, $\mathsf{w}_\tau^* = \mathbf{w}_\tau^*$ and $g := r$, and note that

$$\delta_\tau(\mathbf{w}) = \Delta_\tau(\mathbf{w}) - \tfrac{\mu_{\tau+1}}{2\pi_\tau}(\|\mathbf{w}\|_2^2 - \|\mathbf{w}_{\tau+1}\|_2^2). \tag{83}$$

To see that $\Delta_\tau$ is the Bregman divergence of $r_\tau$, we apply the (sub)differential optimality condition to

$$\mathbf{w}_{\tau+1} := P_r^{1/\mu_{\tau+1}}(\pi_\tau \mathbf{w}_{\tau+1}^*/\mu_{\tau+1}) = \arg\min_{\mathbf{w}} \ \tfrac{1}{2}\|\mathbf{w} - \pi_\tau\mathbf{w}_{\tau+1}^*/\mu_{\tau+1}\|_2^2 + \tfrac{1}{\mu_{\tau+1}}r(\mathbf{w}), \tag{84}$$

so that

$$\mathbf{w}_{\tau+1}^* \in \tfrac{\mu_{\tau+1}}{\pi_\tau}\mathbf{w}_{\tau+1} + \tfrac{1}{\pi_\tau}\nabla r(\mathbf{w}_{\tau+1}) = \nabla r_\tau(\mathbf{w}_{\tau+1}) \tag{85}$$

and hence

$$\Delta_\tau(\mathbf{w}) = r_\tau(\mathbf{w}) - r_\tau(\mathbf{w}_{\tau+1}) - \langle \mathbf{w} - \mathbf{w}_{\tau+1}, \nabla r_\tau(\mathbf{w}_{\tau+1})\rangle. \tag{86}$$

Clearly, $r_\tau$ is convex if $r$ is convex, in which case $\Delta_\tau \geq 0$. $\square$

**Theorem 5.1.** Fix any $\mathbf{w}$, the iterates in (31) satisfy:

$$\sum_{\tau=s}^t \eta_\tau[\langle \mathbf{w}_\tau - \mathbf{w}, \widetilde{\nabla}\ell(\mathbf{w}_\tau)\rangle + r(\mathbf{w}_\tau) - r(\mathbf{w})] \leq \Delta_{s-1}(\mathbf{w}) - \Delta_t(\mathbf{w}) + \sum_{\tau=s}^t \Delta_\tau(\mathbf{w}_\tau), \tag{32}$$

where $\Delta_\tau(\mathbf{w}) := r_\tau(\mathbf{w}) - r_\tau(\mathbf{w}_{\tau+1}) - \langle \mathbf{w} - \mathbf{w}_{\tau+1}, \mathbf{w}_{\tau+1}^* \rangle$ is the Bregman divergence induced by the (possibly nonconvex) function $r_\tau(\mathbf{w}) := \frac{1}{\pi_\tau}r(\mathbf{w}) + \frac{1}{2}\|\mathbf{w}\|_2^2$.

*Proof.* Simply set $\mu_t = \pi_{t-1}$ in Theorem A.3 above. $\qquad\square$

**Corollary 5.2.** For convex $\ell$ and any $\mathbf{w}$, the iterates in (31) satisfy:

$$\min_{\tau=s,\ldots,t} \; \mathbf{E}[f(\mathbf{w}_\tau) - f(\mathbf{w})] \leq \tfrac{1}{\sum_{\tau=s}^t \eta_\tau} \cdot \mathbf{E}\Big[\Delta_{s-1}(\mathbf{w}) - \Delta_t(\mathbf{w}) + \sum_{\tau=s}^t \Delta_\tau(\mathbf{w}_\tau)\Big]. \qquad (33)$$

If r is also convex, then

$$\min_{\tau=s,\ldots,t} \; \mathbf{E}[f(\mathbf{w}_\tau) - f(\mathbf{w})] \leq \tfrac{1}{\sum_{\tau=s}^t \eta_\tau} \cdot \mathbf{E}\Big[\Delta_{s-1}(\mathbf{w}) + \sum_{\tau=s}^t \tfrac{\eta_\tau^2}{2}\|\widetilde{\nabla}\ell(\mathbf{w}_\tau)\|_2^2\Big], \qquad (34)$$

and

$$\mathbf{E}\big[f(\bar{\mathbf{w}}_t) - f(\mathbf{w})\big] \leq \tfrac{1}{\sum_{\tau=s}^t \eta_\tau} \cdot \mathbf{E}\Big[\Delta_{s-1}(\mathbf{w}) + \sum_{\tau=s}^t \tfrac{\eta_\tau^2}{2}\|\widetilde{\nabla}\ell(\mathbf{w}_\tau)\|_2^2\Big], \qquad (35)$$

where $\mathbf{w}_t = \frac{\sum_{\tau=s}^t \eta_\tau \mathbf{w}_\tau}{\sum_{\tau=s}^t \eta_\tau}$.

*Proof.* We first apply the expectation with respect to random sampling to (32) which reduces the left hand side to

$$\mathbf{E}\left[\sum_{\tau=s}^t \eta_\tau[\langle \mathbf{w}_\tau - \mathbf{w}, \widetilde{\nabla}\ell(\mathbf{w}_\tau)\rangle + r(\mathbf{w}_\tau) - r(\mathbf{w})]\right] = \mathbf{E}\left[\sum_{\tau=s}^t \eta_\tau[\langle \mathbf{w}_\tau - \mathbf{w}, \nabla\ell(\mathbf{w}_\tau)\rangle + r(\mathbf{w}_\tau) - r(\mathbf{w})]\right] \tag{87}$$

$$\geq \mathbf{E}\left[\sum_{\tau=s}^t \eta_\tau[\ell(\mathbf{w}_\tau) - \ell(\mathbf{w}) + r(\mathbf{w}_\tau) - r(\mathbf{w})]\right] \tag{88}$$

$$= \mathbf{E}\left[\sum_{\tau=s}^t \eta_\tau[f(\mathbf{w}_\tau) - f(\mathbf{w})]\right], \tag{89}$$

where we used the convexity of $\ell$. We then obtain (33) by using

$$\min_{\tau=s,\ldots,t} \; \mathbf{E}\left[f(\mathbf{w}_\tau) - f(\mathbf{w})\right] \leq \frac{1}{\sum_{\tau=s}^t \eta_\tau} \mathbf{E}\left[\sum_{\tau=s}^t \eta_\tau[f(\mathbf{w}_\tau) - f(\mathbf{w})]\right]. \tag{90}$$

The right-hand sides of (34) and (35) are obtained by setting $\mu_\tau = \pi_{\tau-1}$ and upper bounding the Bregman divergence:

$$\Delta_\tau(\mathbf{w}_\tau) = r_\tau(\mathbf{w}_\tau) - r_\tau(\mathbf{w}_{\tau+1}) - \langle \mathbf{w}_\tau - \mathbf{w}_{\tau+1}, \mathbf{w}_{\tau+1}^*\rangle \tag{91}$$

$$= r_\tau^*(\mathbf{w}_{\tau+1}^*) - r_\tau^*(\mathbf{w}_\tau^*) - \langle \mathbf{w}_{\tau+1}^* - \mathbf{w}_\tau^*, \mathbf{w}_\tau\rangle \quad \text{(by duality of Bregman divergence)} \tag{92}$$

$$\leq \frac{1}{2}\|\mathbf{w}_\tau^* - \mathbf{w}_{\tau+1}^*\|_2^2 \quad \text{(by 1-smoothness of r}^*\text{)} \tag{93}$$

$$= \frac{\eta_t^2}{2}\|\widetilde{\nabla}\ell(\mathbf{w}_t)\|_2^2, \tag{94}$$

where we have used the well-known fact that the Fenchel conjugate of a $(1/L)$-strongly convex function is $L$-smooth (in our case $L = 1$). Lastly, the left-hand side of (35) is obtained by applying the convexity of $f$:

$$\left[\sum_{\tau=s}^t \eta_\tau\right] f(\bar{\mathbf{w}}_t) \leq \sum_{\tau=s}^t \eta_\tau f(\mathbf{w}_\tau), \quad \text{where} \quad \bar{\mathbf{w}}_t = \frac{\sum_{\tau=s}^t \eta_\tau \mathbf{w}_\tau}{\sum_{\tau=s}^t \eta_\tau}. \tag{95}$$

$\qquad\square$

### A.3.1 Discussion of Theorem A.3

Recall the Bregman divergence from Theorem A.3:

$$\Delta_\tau(\mathbf{w}) = \mathsf{r}_\tau(\mathbf{w}) - \mathsf{r}_\tau(\mathbf{w}_{\tau+1}) - \left\langle \mathbf{w} - \mathbf{w}_{\tau+1}, \mathbf{w}_{\tau+1}^* \right\rangle, \quad \mathsf{r}_\tau(\mathbf{w}) = \tfrac{1}{\pi_\tau}\mathsf{r}(\mathbf{w}) + \tfrac{\mu_{\tau+1}}{2\pi_\tau}\|\mathbf{w}\|_2^2. \tag{96}$$

When $\mathsf{r}$ is $\sigma_0$-strongly convex, $\mathsf{r}_\tau$ is $\frac{\sigma_0+\mu_{\tau+1}}{\pi_\tau}$-strongly convex, and hence

$$\Delta_\tau(\mathbf{w}_\tau) \le \tfrac{\pi_\tau}{2(\sigma_0+\mu_{\tau+1})}\|\mathbf{w}_{\tau+1}^* - \mathbf{w}_\tau^*\|_2^2 = \tfrac{\pi_\tau \eta_\tau^2}{2(\sigma_0+\mu_{\tau+1})}\|\widetilde{\nabla}\ell(\mathbf{w}_\tau)\|_2^2, \tag{97}$$

where we used the duality of the Bregman divergence and the smoothness of $\mathsf{r}_\tau^*$. Dividing both sides of (81) by $\sum_{\tau=s}^t \eta_\tau$ we obtain the upper bound:

$$\mathrm{UB} := \frac{\frac{\pi_{s-1}}{\sigma_0+\mu_s}\|\mathbf{w}^* - \mathbf{w}_s^*\|_2^2 + \sum_{\tau=s}^t \left[\frac{\pi_\tau \eta_\tau^2}{\sigma_0+\mu_{\tau+1}}\|\widetilde{\nabla}\ell(\mathbf{w}_\tau)\|_2^2 + (\frac{\mu_{\tau+1}}{\pi_\tau} - \frac{\mu_\tau}{\pi_{\tau-1}})(\|\mathbf{w}\|_2^2 - \|\mathbf{w}_\tau\|_2^2)\right]}{2\sum_{\tau=s}^t \eta_\tau}, \tag{98}$$

where $\mathbf{w}^* := \nabla \mathsf{r}_{s-1}(\mathbf{w})$ and we have dropped the non-positive term $-\Delta_t(\mathbf{w})$. Suppose[8] $\frac{\mu_{t+1}}{\pi_t}$ is non-decreasing w.r.t. $t$, we can thus drop some non-positive terms to further simplify:

$$\mathrm{UB} \le \frac{\frac{\pi_{s-1}}{\sigma_0+\mu_s}\|\mathbf{w}^* - \mathbf{w}_s^*\|_2^2 + \frac{\mu_{t+1}}{\pi_t}\|\mathbf{w}\|_2^2 + \sum_{\tau=s}^t \left[\frac{\pi_\tau \eta_\tau^2}{\sigma_0+\mu_{\tau+1}}\|\widetilde{\nabla}\ell(\mathbf{w}_\tau)\|_2^2\right]}{2\sum_{\tau=s}^t \eta_\tau}, \tag{99}$$

To minimize the upper bound, we consider two cases:

- $\sigma_0 = 0$, in which case let us choose $\mu_{\tau+1} = c\sqrt{\eta_\tau \pi_\tau} = c\sqrt{\lambda_\tau}$ (recall that we reparameterized $\eta_\tau = \lambda_\tau/\pi_\tau$ from GCG), where $c$ is an absolute constant. Then, the upper bound reduces to

$$\frac{\frac{\pi_{s-1}}{\mu_s}\|\mathbf{w}^* - \mathbf{w}_s^*\|_2^2 + \frac{c\eta_t}{\sqrt{\lambda_t}}\|\mathbf{w}\|_2^2 + \sum_{\tau=s}^t \eta_\tau\sqrt{\lambda_\tau}\|\widetilde{\nabla}\ell(\mathbf{w}_\tau)\|_2^2/c}{2\sum_{\tau=s}^t \eta_\tau}, \tag{100}$$

  where recall that $\pi_t = \frac{1}{1+\sum_{\tau=1}^t \eta_\tau}$. When the gradient $\nabla\ell$ is bounded, we may choose

$$\lambda_t = \frac{\eta_t}{1+\sum_{\tau=1}^t \eta_\tau} = O(1/t) \text{ and } \frac{1}{\sum_{\tau=1}^t \eta_\tau} = O(1/\sqrt{t}) \tag{101}$$

  so that the upper bound diminishes[9] at the rate of $O(1/\sqrt{t})$. This result makes intuitive sense, since we know the optimal step size $\lambda_t$ in GCG is $\Theta(1/t)$. Also, it reveals that the smoothing parameter $\mu_t = O(1/\sqrt{t})$, matching the rate of the upper bound (i.e., progress on the original problem). Interestingly, the choice in (101) can be realized in multiple ways. In fact, $\eta_t = \eta_0 t^{-p}$ for any $p \in [0, \frac{1}{2}]$ suffices, In particular, choosing $p = 0$ leads to the constant step size $\eta_t \equiv \eta_0$. However, note that these choices are in some sense equivalent, since they all lead to $\lambda_t = O(1/t)$ and $\mu_t = O(1/\sqrt{t})$ hence the underlying GCG progresses similarly.

- $\sigma_0 > 0$, in which case we can further relax the upper bound to:

$$\mathrm{UB} \le \frac{\frac{\pi_{s-1}}{\sigma_0+\mu_{s-1}}\|\mathbf{w}^* - \mathbf{w}_s^*\|_2^2 + \frac{\mu_{t+1}}{\pi_t}\|\mathbf{w}\|_2^2 + \sum_{\tau=s}^t \left[\frac{\pi_\tau \eta_\tau^2}{\sigma_0}\|\widetilde{\nabla}\ell(\mathbf{w}_\tau)\|_2^2\right]}{2\sum_{\tau=s}^t \eta_\tau}, \tag{102}$$

  and now we may choose $\mu_{\tau+1} = c\eta_\tau \pi_\tau = c\lambda_\tau$, which approaches 0 significantly faster than before. With this choice, the upper bound reduces to

$$\frac{\frac{\pi_{s-1}}{\sigma_0+\mu_{s-1}}\|\mathbf{w}^* - \mathbf{w}_s^*\|_2^2 + c\eta_t\|\mathbf{w}\|_2^2 + \sum_{\tau=s}^t \eta_\tau\lambda_\tau\left[\|\widetilde{\nabla}\ell(\mathbf{w}_\tau)\|_2^2/\sigma_0\right]}{2\sum_{\tau=s}^t \eta_\tau}. \tag{103}$$

  Again, we may set $\lambda_t$ as in (101), and we can choose constant $\eta_t \equiv \eta_0$. The major difference is that we may now decrease the smoothing parameter $\mu_t$ much more aggressively.

---

[8]This assumption can be easily satisfied. In fact, we can just set $\mu_{t+1} = \pi_t$ (as in the main paper), which would simplify the discussion quite a bit.

[9]Note that by choosing $s \propto t$, the averaged sequence $\frac{\sum_{\tau=s}^t \eta_\tau \sqrt{\lambda_\tau}}{\sum_{\tau=s}^t \eta_\tau} \le \max_{\tau=s,\dots,t} \sqrt{\lambda_\tau}$ diminishes similarly in order as $\sqrt{\lambda_t}$.

# B    Implementation and Experiment Details

## B.1    CIFAR-10

### B.1.1    Model and Quantization Details

Similar to Bai et al. [5], we use ResNets for which we quantize all weights. BatchNormalization layers and activations are kept at full precision. The basic ResNet implementation is taken from https://github.com/akamaster/pytorch_resnet_cifar10/blob/master/resnet.py.

### B.1.2    Data Augmentation

We follow the data augmentation strategy from Bai et al. [5]: padding by four pixels on each side, randomly cropping to 32-by-32 pixels, horizontally flipping with probability one half. Finally, the images are normalized by subtracting $(0.4914, 0.4822, 0.4465)$ and subsequently dividing by $(0.247, 0.243, 0.261)$.

### B.1.3    Pretrained Full Precision Model Setup

We pretrain two single full precision models for 200 epochs using the standard optimization setup: SGD with 0.9 momentum and $1\mathrm{e}{-}4$ weight decay. The initial learning rate is $0.1$ which is multiplied by 0.1 at epoch 100 and 150. We use batch size 128.

### B.1.4    Fine-Tuning Setup

All methods are initialized with the last checkpoint of the full precision models.

For BinaryConnect, we train with the recommended strategy from Courbariaux et al. [11]: Adam with learning rate 0.01 and multiplication of the learning rate by $0.1$ at epoch 81 and 122. For ProxQuant we use Adam with fixed learning rate of 0.01 as in Bai et al. [5]. We did not perform hyperparameter search over the optimization setup for ProxConnect but rather just used the optimization setup from BinaryConnect as the methods are very similar. We use batch size 128.

Similar to Bai et al. [5], we perform a hard quantization at epoch 200: all weights are projected to their closet quantization points. As in Bai et al. [5], we then train BatchNormalization layers for another 100 epochs.

### B.1.5    End-To-End Setup

For simplicity and to avoid an expensive hyperparameter search, we followed the above full precision optimization setup for all methods. Similar to the fine-tuning setup, we perform a hard quantization at epoch 200 and keep training BatchNormalization layers for another 100 epochs. We use batch size 128.

### B.1.6    Compute and Resources

We run all CIFAR-10 experiments on our internal cluster with GeForce GTX 1080 Ti. We use one GPU per experiment. The total amount of GPU hours is summarized in Table 5.

Table 5: Compute for CIFAR-10 experiments measured in hours per single GeForce GTX 1080 Ti GPU.

| Architecture | Run time | # of total experiments | Total run time |
|---|---|---|---|
| ResNet20 | 1.5 | 181 | 271.5 |
| ResNet56 | 3 | 181 | 543 |

### B.1.7    Additional Results

As mentioned in Section 6.1, we perform a small grid search over $\rho_0$. After an initial exploration stage, we found good regions of $\rho_0$ for ProxQuant and ProxConnect. Since ProxQuant and

reverseProxConnect are quite similar, in particular in the small $\rho$ regime, we simply used the $\rho_0$ from ProxQuant for reverseProxConnect. See below the results for all $\rho_0$ settings.

ProxConnect is reasonably stable with respect to the choice of $\rho_0$ for both fine-tuning and end-to-end training. ProxQuant and reverseProxConnect, on the other hand, are very sensitive to the choice of $\rho_0$ for end-to-end training. ProxConnect reduces to BinaryConnect for large $\rho$, and therefore it is in line with our experiments that ProxConnect should be stable with respect to $\rho_0$, particularly choosing $\rho_0$ too large should not be an issue.

Table 6: Additional results for fine-tuning ProxQuant.

| Model | Quantization | $\rho_0 = 5e{-}7$ | $\rho_0 = 1e{-}6$ | $\rho_0 = 2e{-}6$ |
|---|---|---|---|---|
| ResNet20 | Binary | 89.68 (0.10) | **89.94** (0.10) | 89.34 (0.33) |
| | Ternary | 91.03 (0.07) | **91.46** (0.05) | 91.11 (0.10) |
| | Quaternary | 91.10 (0.06) | 91.13 (0.18) | **91.43** (0.17) |
| ResNet56 | Binary | 92.25 (0.08) | **92.33** (0.06) | 92.16 (0.14) |
| | Ternary | 92.52 (0.90) | **93.07** (0.02) | 92.71 (0.16) |
| | Quaternary | 92.49 (0.03) | **92.82** (0.13) | 92.80 (0.15) |

Table 7: Additional results for fine-tuning reverseProxConnect.

| Model | Quantization | $\rho_0 = 5e{-}7$ | $\rho_0 = 1e{-}6$ | $\rho_0 = 2e{-}6$ |
|---|---|---|---|---|
| ResNet20 | Binary | 89.88 (0.23) | **89.98** (0.17) | 89.91 (0.24) |
| | Ternary | 91.30 (0.05) | 91.36 (0.03) | **91.47** (0.15) |
| | Quaternary | 91.05 (0.12) | **91.43** (0.05) | 91.42 (0.07) |
| ResNet56 | Binary | 92.18 (0.06) | 92.35 (0.02) | **92.47** (0.29) |
| | Ternary | **92.84** (0.09) | 92.85 (0.12) | 92.84 (0.17) |
| | Quaternary | 92.66 (0.20) | 92.88 (0.17) | **92.91** (0.22) |

Table 8: Additional results for fine-tuning ProxConnect.

| Model | Quantization | $\rho_0 = 5e{-}3$ | $\rho_0 = 1e{-}2$ | $\rho_0 = 2e{-}2$ |
|---|---|---|---|---|
| ResNet20 | Binary | 89.63 (0.26) | 90.29 (0.07) | **90.31** (0.21) |
| | Ternary | 91.31 (0.07) | **91.37** (0.18) | 91.13 (0.27) |
| | Quaternary | 91.62 (0.21) | 91.55 (0.10) | **91.81** (0.14) |
| ResNet56 | Binary | 92.41 (0.13) | 92.62 (0.09) | **92.65** (0.16) |
| | Ternary | 93.17 (0.04) | **93.25** (0.12) | 93.22 (0.06) |
| | Quaternary | 93.41 (0.11) | **93.42** (0.12) | 93.28 (0.06) |

Table 9: Additional results end-to-end training ProxQuant.

| Model | Quantization | $\rho_0 = 1e{-}7$ | $\rho_0 = 1e{-}6$ | $\rho_0 = 1e{-}5$ |
|---|---|---|---|---|
| ResNet20 | Binary | **81.59** (0.75) | 81.49 (0.41) | 71.90 (0.75) |
| | Ternary | 28.22 (1.70) | 41.08 (2.95) | **47.98** (1.06) |
| | Quaternary | 84.58 (0.15) | **85.29** (0.07) | 75.08 (0.16) |
| ResNet56 | Binary | **86.13** (1.71) | 80.25 (0.51) | 68.31 (2.21) |
| | Ternary | 21.93 (2.43) | 41.11 (2.12) | **50.54** (3.01) |
| | Quaternary | **87.81** (1.30) | 83.57 (1.70) | 72.58 (2.23) |

Table 10: Additional results end-to-end training reverseProxConnect.

| Model | Quantization | $\rho_0 = 1e{-}7$ | $\rho_0 = 1e{-}6$ | $\rho_0 = 1e{-}5$ |
|---|---|---|---|---|
| ResNet20 | Binary | **81.82** (0.32) | 80.84 (0.40) | 72.10 (1.01) |
| | Ternary | 26.49 (2.82) | 40.78 (0.39) | **47.17** (1.94) |
| | Quaternary | **85.05** (0.27) | 84.82 (0.32) | 75.61 (0.54) |
| ResNet56 | Binary | **86.25** (1.50) | 81.58 (0.92) | 67.53 (2.74) |
| | Ternary | 21.51 (1.12) | **42.95** (1.57) | 36.34 (18.68) |
| | Quaternary | **87.30** (1.02) | 84.72 (1.31) | 73.36 (1.36) |

Table 11: Additional results end-to-end training ProxConnect.

| Model | Quantization | $\rho_0 = 5e{-}3$ | $\rho_0 = 1e{-}2$ | $\rho_0 = 2e{-}2$ |
|---|---|---|---|---|
| ResNet20 | Binary | 89.72 (0.13) | **89.92** (0.26) | 89.65 (0.15) |
| | Ternary | **84.09** (0.16) | 83.54 (0.36) | 82.84 (0.34) |
| | Quaternary | **90.17** (0.14) | 90.12 (0.33) | 89.91 (0.09) |
| ResNet56 | Binary | **91.26** (0.59) | 90.45 (0.83) | 89.29 (0.45) |
| | Ternary | **84.36** (0.75) | 83.46 (1.14) | 82.54 (1.38) |
| | Quaternary | 91.00 (0.50) | 90.76 (0.54) | **91.70** (0.14) |

**B.2    ImageNet**

**B.3    Model and Quantization Details**

We use ResNet18 as our model, for which we quantize all weights except for the first convolutional layer and the last fully-connected layer. Other components such as BatchNormalization layers, activations and biases are kept at full-precision. The ResNet implementation is borrowed from PyTorch's torchvision package: `https://github.com/pytorch/vision/blob/882e11db8138236ce375ea0dc8a53fd91f715a90/torchvision/models/resnet.py`.

**B.4    Data Augmentation**

Common data augmentation strategy is followed from `https://github.com/pytorch/examples/blob/c002856901eaf9be112feb9b14a9d5c3e779da74/imagenet/main.py#L204-L231`. At training time, the images are randomly resized and cropped to 224-by-224 pixels, followed by a random horizontal flip. At inference time, images are first resized to 256-by-256 pixels, then center cropped to 224-by-224 pixels. Finally, the images are normalized by subtracting $(0.485, 0.456, 0.406)$ and subsequently dividing by $(0.229, 0.224, 0.224)$.

**B.5    Pretrained Full Precision Model Setup**

We use the ResNet18 checkpoint provided by torchvision as our pretrained model: `https://download.pytorch.org/models/resnet18-f37072fd.pth`.

**B.6    Fine-Tuning Setup**

All fine-tuned models are initialized from the pretrained model described in the previous section.

As advised by Alizadeh et al. [2], we use Adam as the optimizer to fine-tune the models. We use Adam with default parameters, initial learning rate $1e{-}4$ and batch size 256. The models are fine-tuned for 50 epochs. The learning rate is divided by 10 at epoch 15 and 30. We perform hard quantization at epoch 45 and train the remaining full-precision layers for the last 5 epochs, mostly to let BatchNormalization layers stabilize.

**B.7    End-To-End Setup**

For the end-to-end setup, the models are trained from scratch mimicking the training setup of the full-precision pretrained model. The quantized ResNet18 is trained for 90 epochs using SGD with a starting learning rate of $0.1$, momentum of $0.9$ and weight decay of $1e{-}4$. The learning rate is multiplied by $0.1$ at epoch 30 and 60. We use batch size 256. We perform hard quantization at epoch 80 and train the remaining full-precision layers for the last 10 epochs.

**B.8    Compute and Resources**

We run all experiments on our internal cluster with Tesla V100. We use one GPU per experiment. The total amount of GPU hours is summarized in Table 12.

Table 12: Compute for ImageNet experiments measured in hours per single Tesla V100 GPU.

| Approach | Run time | # of total experiments | Total run time |
|---|---|---|---|
| End-To-End | 27 | 9 | 243 |
| Fine-Tuning | 12 | 9 | 108 |