# OpenReview forum: "Demystifying and Generalizing BinaryConnect"
_NeurIPS.cc/2021/Conference — NeurIPS 2021 Poster_

### Official Review · Reviewer_bjiL · 2021-06-29

**Rating:** 7
**Confidence:** 3

**Summary:**

This paper seeks to give a theoretical grounding to the study of the quantisers used in the training of neural networks. The authors use proximal maps as the foundation to construct quantisers, and demonstrate that a host of known quantisation techniques can be cast in this manner.

The authors show that BinaryConnect is a special case of the GCG algorithm, and use this as motivation to construct their own more general version - Proximal Connect - which contains BinaryConnect as a special case.

Results are shown on classification of CIFAR-10, and demonstrate that the proposed method improves performance on this task over BinaryConnect and ProxQuant (the main methods this work focuses on).

**Ethical Concerns:**

I don't see any ethical concerns.

**Limitations And Societal Impact:**

I do not see any societal impact of the work, since it is a primarily theoretical work focused on quantisation.

**Main Review:**

This work has two types of contribution - the theoretical understanding given about already existing quantisers, and the proposal of a new quantiser (PC) which the authors show has advantages over existing techniques.

I think in general the theoretical contributions are valuable to the community, since the quantisation techniques used in training quantised neural networks are somewhat poorly understood. Although the casting of the quantisation problem in the language of proximal maps is not novel, the relation to the various known quantisers in ML is useful.

The authors then show that BinaryConnect (the main quantiser used for binary weights) is a special case of the GCG algorithm. It was already known that BC was a special case of the regularised dual averaging problem, but the GCG does generalise it further. It is not clear to me exactly how valuable this realisation is. At the least, the authors use it to motivate the construction of their PC quantiser, and perhaps there could be more developments that are enabled by this knowledge.

The new PC quantiser is then demonstrated to improve results on a fine-tuning task for image classification. Although this is a relatively limited evaluation (there is only one experiment, CIFAR-10, considered) there is enough theoretical meat to justify this in my eyes. The improvements are seemingly significant, in that BC and PQ each have failure cases in the given experiments, that PC does not have. Although these failure cases (for example end-to-end training of ternary weights for PQ) could be resolved by other existing techniques, it seems that the entire points of this work is to try and provide a more solid and reliable quantiser by examining overarching theory, rather than developing brittle quantisers for special cases. The PC quantiser itself seems to be relatively straightforward, after all is said and done, in that it is just a linear piecewise function of varying slopes and level intervals around quantisation points. I would say that simplicity is a good thing though, if such a technique is going to be adopted by the community.

I also find in general the work very well-written, although at points it can be dense in notation and a bit tricky to follow through the more involved sections.

Some small points:
 * the authors mention that stochastic quantisation can be cast in the proximal mapping framework. Do they have any thoughts about how the class of methods for training stochastic quantisers might relate to their framework? For example [1] and [2].
* the authors talk about the reverse BC and reverse PC algorithms, giving experimental results for the latter. Although this is clearly a scheme that we can write down, and it is motivated by the PQ algorithm, it does not seem intuitive to me. Why would we want to update the continuous parameters starting from the locations of the current quantised parameters? Perhaps this is just a gap in my optimisation intuition.

[1] LEARNING DISCRETE WEIGHTS USING THE LOCAL REPARAMETERIZATION TRICK
Oran Shayer et al. ICLR 2018

[2] Probabilistic Binary Neural Networks
Jorn W.T. Peters et al. 2018

**Time Spent Reviewing:**

4

---

> ### Author Response · Authors · 2021-08-10
> **Thank you for your questions and feedback**
>
> Thank you for the detailed review and thoughtful feedback. Below we address specific questions and comments.
>
> **Q: It was already known that BC was a special case of the regularised dual averaging problem, but the GCG does generalise it further. It is not clear to me exactly how valuable this realisation is.**
>
> A: Thank you for bringing this up. In our opinion, this connection is valuable in the following ways:
> - We believe our explanation of DA using GCG is of independent interest to the ML community. For example, the excellent book of Bubeck (mentioned by Reviewer 5Fsk) touched on both DA and conditional gradient (i.e., Frank--Wolfe) but never connected the two. To give another example, Nesterov in his original work ([28]) motivated DA by the desire of assigning bigger weights to more recent iterates, whereas conventional subgradient algorithms with diminishing step size "counter-intuitively" assign smaller weights instead. Based on our explanation, we can now see clearly why that is possible: DA solves an (approximate) *smooth* dual problem, hence we can afford to use a constant (rather than diminishing) step size (i.e., equal weights for all iterates). This view is also useful in understanding what BC really aims to solve even in the convex setting.
>
> - GCG is an extremely simple algorithm with a very short convergence proof. It is of educational value to identify DA as a special case of GCG: students are excused of learning another proof and gain more understanding of both algorithms.
>
> - Lastly, there has been a lot of recent works that extend GCG along many different axes (while relatively few for DA and BC), see below for some sample references. So, with our results, it may now become plausible to transfer these advances on GCG to BC and NN quantization. We mentioned this in our conclusion and it is an active direction that we are currently pursuing.
>
> References:
> - Leonard Berrada, Andrew Zisserman and M. Pawan Kumar (2019). "Deep Frank-Wolfe For Neural Network Optimization". ICLR.
>
> - Mingrui Zhang, Lin Chen, Aryan Mokhtari, Hamed Hassani and Amin Karbasi (2020). "Quantized Frank-Wolfe: Faster Optimization, Lower Communication, and Projection Free". AISTATS.
>
> - Pavel Dvurechensky, Petr Ostroukhov, Kamil Safin, Shimrit Shtern, Mathias Staudigl (2020). "Self-Concordant Analysis of Frank-Wolfe Algorithms". ICML.
>
> **Q: The PC quantiser itself seems to be relatively straightforward, after all is said and done, in that it is just a linear piecewise function of varying slopes and level intervals around quantisation points. I would say that simplicity is a good thing though, if such a technique is going to be adopted by the community.**
>
> A: We would like to point out that our theory works for *any* proximal quantizer, and includes previous non-piecewise attempts (e.g. [1]) as special cases. We chose a piecewise quantizer in our experiments precisely because of its simplicity (along with the ease to compare with prior work): based on the current available results, we do not have any theoretical reason (yet) to prefer any specific quantizer so simplicity triumphs in our final design. We greatly appreciate the reviewer's recognition of simplicity.
>
> **Q: The authors mention that stochastic quantisation can be cast in the proximal mapping framework. Do they have any thoughts about how the class of methods for training stochastic quantisers might relate to their framework? For example (Shayer et al., 2018) and (Peters and Welling, 2018).**
>
> A: Thank you for bringing these interesting references to our attention; we will gladly cite and discuss them. Let us clarify what we mean with stochastic quantization in our work: ProxConnect is able to employ stochastic quantization through a mixture of proximal maps as described in Equation 9. In each step, a component $i$ can be sampled from the categorical distribution with parameter $\\{\\alpha_j\\}_{j=1}^k$ and the corresponding proximal map $\\mathsf{P}_i$ would be applied to the weights in this particular iteration.
>
> The suggested references, on the other hand, incorporate stochasticity by sampling weights (rather than proximal maps as we do) from a normal distribution with learnable parameters. These correspond to probabilistic networks while we focus on deterministic networks.
>
> Let us now explain the connections and subtleties. For simplicity we restrict ourselves to (Shayer et al., 2018) and the binary quantization setting. Shayer et al. (2018) sample each weight from a univariate normal distribution with mean $2p -1$ and variance $4p(1-p)$. Here $p \in [0,1]$ are the learned parameters. The arising transformation (from parameter to weight) can be written as $\\mathsf{T}_\\varepsilon(p) = 2p - 1 + 2 \\sqrt{p(1-p)} \\varepsilon$, where $\\varepsilon \sim \\mathcal{N}(0, 1)$. If $p$ were fixed, ProxConnect could realize this setup by having a continuum of mixtures with corresponding constant-valued proximal maps $\\mathsf{P}_u \\equiv 2 p - 1 + 2 \\sqrt{p (1-p)} \\Phi^{-1}(u)$, where $\\Phi$ is the CDF of a standard normal and $u$ is uniform over [0,1]. In practice, however, the parameters $p$ are changing in every step so our Theorem 3.2 does not readily apply.
>
> In principle, we believe it is possible to extend the previous argument to also cover probabilistic networks, although this is not our focus here and is best done in future work. We thank you for this very inspiring comment.
>
> **Q: The authors talk about the reverse BC and reverse PC algorithms, giving experimental results for the latter. Although this is clearly a scheme that we can write down, and it is motivated by the PQ algorithm, it does not seem intuitive to me. Why would we want to update the continuous parameters starting from the locations of the current quantised parameters?**
>
> A: We agree with the reviewer that reversePC (and reverseBC as a special case) may not be intuitive. The reasons to include them are:
> - As shown in Section 2, reverseBC is discovered by writing down common quantization algorithms in a unified form. For completeness, we wanted to try out the last possibility (out of the 4 combinations), even though it is less intuitive.
> - The way we explain these algorithms also makes it extremely easy to implement and compare all of them in a common code base. To our surprise, reversePC, the generalization of reverseBC, actually performed reasonably well and we considered it worthwhile to report these experimental results, in hopes that it may inspire further research.
> - reversePC is not completely without merit: it evaluates the gradient at the  continuous weights $\\mathbf{w}_t^\star$ and hence is able to exploit richer landscape of the loss. Even when stuck at a fixed discrete weight $\\mathbf{w}_t$, reversePC may still accumulate sizable updates (as long as the step size and the gradient remain sufficiently large) to allow it to eventually jump out of $\\mathbf{w}_t$: note that the continuous weight $\\mathbf{w}_t^\\star$ still gets updated. Finally, we note that fixed points of reversePC, when exist, satisfy:
>
> $\\mathbf{w}^\\star = \\mathsf{P}^{\\eta}_{\\mathsf{r}}(\\mathbf{w}^\\star) - \\eta \\nabla f(\\mathbf{w}^\\star) \\iff \\mathbf{w}^\\star = (\\mathsf{id} + \\eta \\nabla f)^{-1} (\\mathsf{id} + \\eta \\partial \\mathsf{r})^{-1} \\mathbf{w}^\\star$,
>
> where for simplicity we use a fixed step size $\\eta$ and assume $\\mathsf{r}$ is convex (so that $\\mathsf{P}^{\\eta}_{\\mathsf{r}} = (\\mathsf{id} + \\eta \\partial \\mathsf{r})^{-1}$). The operator on the right-hand side is known as the backward-backward update (as opposed to the forward-backward update in ProxQuant), and it is known that when $\\eta \\to 0$ slowly, backward-backward update converges to a stationary point. These details can be derived from the classic work of Passty. Thus, despite of our current limited understanding of reversePC, there is some reason and empirical evidence to believe it might still be interesting.
>
> Reference:
> Passty, Gregory B. (1979). ["Ergodic convergence to a zero of the sum of monotone operators in Hilbert space"](https://doi.org/10.1016/0022-247X(79)90234-8). Journal of Mathematical Analysis and Applications, vol. 72, no. 2, pp. 383–390.
>
> We would be happy to include the above discussions in a final revision.

---

### Official Review · Reviewer_Ub8c · 2021-07-17

**Rating:** 7
**Confidence:** 3

**Summary:**

This paper proposes a generalized conditional gradient (GCG) framework for training weight-quantized neural networks. The framework extends existing BinaryConnect and ProxQuant. Convergence bound is established for general, non-convex problem. Compared to ProxQuant, the proposed theory supports evolving quantizers. The paper also proposes a family of smooth quantizers, parameterized by the horizontal and vertical shift. The proposed ProxConnect outperforms existing approaches on end-to-end training of ResNets.

**Limitations And Societal Impact:**

This work doesn't have any foreseeable potential negative societal impact.

**Main Review:**

Principled training methods for quantized neural networks is a important problem to study. This paper generalizes existing theory on training weight-quantized neural networks with a generalized conditional gradient (GCG) framework. The unified view of existing methods in Sec. 2 is interesting.

However, I think the contribution of this paper is somewhat incremental, as the convergence is already established for smooth and non-convex problems in Bai et al.. The generalization of dual averaging under the GCG framework is one of the main contribution of this paper. However, the advantage of GCG over DA is not sufficiently demonstrated. There lacks a strong showcase to explain why the general theoretical result (Theorem 5.1) is useful. If the authors want to emphasize time-dependent quantizers, I think more explanations should be included. Can we theoretically justify why the proposed quantizers can achieve better results, e.g., through a better convergence bound?

The paper is somewhat hard to read and understand. I am particularly confused on the role of Sec. 4. If my understanding is correct, in the non-convex setting, the dual problem (17) is only a lower bound of the original problem (14). In this case, why is Theorem 4.1 and Corollary 4.2 useful, as they don't directly discuss the original problem? If these theorems are not directly applicable, I would suggest to put more effort on the more relevant Sec. 5. For example, I think Theorem 5.1 needs more explanations. What does the term Delta_tau looks like if r(w) is an indicator function? Can Theorem 5.1 reduces to Theorem 5.1 in the ProxQuant paper if the quantizer is fixed?

Figure 2 and Table 1 are too messy to read. Please consider replacing the random symbols with text.

Post Rebuttal
====

After reading the authors' response and spent several more hours in reading relevant papers I am convinced that the submission made a non-trivial contribution upon existing works (Li et al., Bai et al.), in the sense that it formally shows the convergence for BinaryConnect for non-convex problems. The natural treatment of increasingly "discrete" proximal map is also interesting. Therefore I increased my rating from 5 to 7 and the confidence from 2 to 3.

**Time Spent Reviewing:**

6

---

> ### Author Response · Authors · 2021-08-10
> **Thank you for your questions and feedback**
>
> Thank you for the detailed review and thoughtful feedback. Below we address specific questions and comments.
>
> **Q: However, I think the contribution of this paper is somewhat incremental, as the convergence is already established for smooth and non-convex problems in Bai et al.. The generalization of dual averaging under the GCG framework is one of the main contribution of this paper. However, the advantage of GCG over DA is not sufficiently demonstrated. There lacks a strong showcase to explain why the general theoretical result (Theorem 5.1) is useful. If the authors want to emphasize time-dependent quantizers, I think more explanations should be included. Can we theoretically justify why the proposed quantizers can achieve better results, e.g., through a better convergence bound?**
>
> A: There seems to be some misunderstanding here. Bai et al. (and others) proved the convergence of ProxQuant (a.k.a., forward-backward splitting) while our contribution is on a related but fundamentally different algorithm (PC/BC). As we mentioned in the introduction (see also [1]), BC, despite of its practical popularity, has largely remained as a training heuristic. Our work put a rigorous justification for BC and led to the novel generalization PC, with any proximal quantizer that one can now effortlessly design. The general Theorem 5.1, as well as its derived Corollary 5.2, is the most general convergence guarantee for PC/BC (to the best of our knowledge). We thank you for suggesting more discussions around Theorem 5.1, which we will follow by rearranging the appendix and incorporating all comments and response here.
>
> The advantage of our GCG explanation is three-fold: (1) It is more intuitive, and its proof is much simpler and extends immediately to the nonconvex setting (in contrast to existing work that had to start from scratch and hence brought in unnecessary complications). (2) It allows us to formally justify the widely-adopted diverging step size $\\nu_t$. (3) As mentioned in our conclusion, it opens the door to transfer many recent advances on GCG to NN quantization (such as the following).
>
> References:
> - Leonard Berrada, Andrew Zisserman and M. Pawan Kumar (2019). "Deep Frank-Wolfe For Neural Network Optimization". ICLR.
> - Mingrui Zhang, Lin Chen, Aryan Mokhtari, Hamed Hassani and Amin Karbasi (2020). "Quantized Frank-Wolfe: Faster Optimization, Lower Communication, and Projection Free". AISTATS.
> - Pavel Dvurechensky, Petr Ostroukhov, Kamil Safin, Shimrit Shtern, Mathias Staudigl (2020). "Self-Concordant Analysis of Frank-Wolfe Algorithms". ICML.
>
> As we tried to highlight throughout the paper, time-dependent quantizers are an important ingredient to the empirical success of many quantization algorithms, such as the ones proposed in [1] and [5]. As we described in Section 3, our proposed quantizer $\\mathsf{L}_{\\rho}^\\varrho$ was inspired by generalizing quantizers from works such as BinaryRelax [39] and ProxQuant [5]. We do not claim that time-dependent quantizers are needed for *any* quantization algorithm, but rather it is the combination of the quantizer and the update algorithm that matters. For example, our proposed ProxConnect reduces to BinaryConnect when using a particular time-independent quantizer ($\\rho = \\varrho = \\infty$). We further illustrated the advantage of the proposed quantizer in Figure 2 (updated for better clarity) and thoroughly verified it in our experiments.
>
> **Q: The paper is somewhat hard to read and understand. I am particularly confused on the role of Sec. 4. If my understanding is correct, in the non-convex setting, the dual problem (17) is only a lower bound of the original problem (14). In this case, why is Theorem 4.1 and Corollary 4.2 useful, as they don't directly discuss the original problem? If these theorems are not directly applicable, I would suggest to put more effort on the more relevant Sec. 5. For example, I think Theorem 5.1 needs more explanations.**
>
> A: We apologize for the somewhat dense presentation in Sections 4 and 5. We will add more explanations of the theorems and bring some of the discussions in the appendix to the main paper. Thank you for these suggestions.
>
> Regarding Section 4, we believe it is an important part of our paper and we will highlight its importance in the introduction of our revision:
> - The discussion of the convex case (Section 4) largely motivated our development of the nonconvex setting (Section 5). It is reassuring that our theoretical results made sense for "simpler" convex functions, and by comparing the two sections we can appreciate the similarities and challenges for dealing with nonconvex functions. Our proofs for Theorem 5.1 and Corollary 5.2 are patterned after those for Theorem 4.1 and Corollary 4.2.
> - We believe our explanation of the dual averaging algorithm using GCG is of independent interest to the ML community, which is best done through convex functions (Section 4). In fact, after communicating with top experts on dual averaging, we were encouraged to publish this result. For example, the excellent book of Bubeck (mentioned by Reviewer 5Fsk) touched both dual averaging and conditional gradient (i.e., Frank--Wolfe) but never connected the two. Theorem 4.1 and Corollary 4.2 also justify our claim that DA is GCG applied to a smoothened dual: they allow us to see clearly what DA really aims to optimize and to recover the exact convergence rates of DA in the convex setting.
>
> **Q: What does the term Delta_tau looks like if r(w) is an indicator function?**
>
> A: When $\\mathsf{r} = \iota_Q$ is an indicator function, $ \Delta_{\tau}(\\mathbf{w}) = \\tfrac{1}{2} \|\|\\mathbf{w} - \\mathbf{w}_{\\tau+1}\|\|_2^2$  if $\\mathbf{w} \in Q$ and $\\infty$ otherwise. We will add the above clarifications to our final revision.
>
> **Q: Can Theorem 5.1 reduces to Theorem 5.1 in the ProxQuant paper if the quantizer is fixed?**
>
> A: Our Theorem 5.1 is about ProxConnect (which includes BC as a special case) while Theorem 5.1 of Bai et al. is about ProxQuant. Although the two algorithms (ProxConnect and ProxQuant) are quite similar in their updates (see Eqs (3) and (5) for comparison), they are not directly comparable. Arguably, establishing convergence bounds for PC/BC (in the nonconvex setting) is significantly harder while the convergence for ProxQuant follows classic work on forward-backward splitting, see e.g.
>
> Hedy Attouch, Jerome Bolte and Benar Fux Svaiter (2013). "Convergence of descent methods for semi-algebraic and tame problems: proximal algorithms, forward-backward splitting, and regularized Gauss-Seidel methods". Mathematical Programming, vol. 137, pp. 91--129.
>
> The proof of ProxQuant failed to explain the practice of using a diverging step size, which is very crucial in obtaining competent experimental results (and is also adopted in ProxQuant's experiments), while our convergence proof fully justifies this practice for ProxConnect and BC.
>
> **Q: Figure 2 and Table 1 are too messy to read. Please consider replacing the random symbols with text.**
>
> A: Thank you for this suggestion. We have incorporated the suggested changes by (a) replacing Figure 2 with an explanation and publishing a slide deck that highlights the differences between the algorithms; (b) modifying Table 1 for improved readability and clearness. See this anonymized [folder](https://drive.google.com/drive/folders/1alxYqAebe11hh5OWd2T6xvLtzzyr3sW0?usp=sharing) for both the slide deck and the updated table.

---

> ### Author Response · Authors · 2021-08-31
> **Post-rebuttal update comment**
>
> We thank the reviewer for their active participation in the review process and appreciate their dedication to spending more time on relevant literature.

---

### Official Review · Reviewer_5Fsk · 2021-07-20

**Rating:** 7
**Confidence:** 4

**Summary:**

The paper presents a refined theoretical understanding of binary connect via relating the dual-averaging method to generalized conditional gradient descent. Then, a general proximal connect algorithm is presented. Experiments on image classification datasets show marginal improvements over the comparable methods.

**Limitations And Societal Impact:**

Adequate

**Main Review:**

## Post-rebuttal Update
I thank the authors for clarifying the questions. I would strongly encourage them to include the discussion related to MD and [1] clearly in the main paper to clarify the main contributions of the paper. Upon this condition, I increase the score to 7.

## Strengths
- The summarization of different quantization methods under one umbrella is interesting and useful.
- Even though there are many attempts to provide a wholesome theory for binary connect previously, this paper refines the understanding by including the annealing hyperparameter into the theoretical framework.
- The connection to generalized conditional gradient descent seems interesting.

## Weaknesses
- Not clear mirror descent (MD) would be a special case of proximal connect: In my understanding, dual averaging is a special case of MD [a] and generalizing dual averaging does not mean MD would be a special case of the proposed method. Furthermore, MD supports nonlinear projections (such as tanh or softmax [1]) whereas the proposed formulation supports piecewise linear projections. So I'm not sure PC would be a generalization of the methods discussed in [1]. Please clarify.

- The main idea of this paper is in Eq. 8 (and Eq. 31, if I understand correctly) where the regularizer $r$ is parametrized as opposed to ProxQuant [5]. May be this interpretation should be mentioned early so as to simplify the understanding of the final PC algorithm.


## References
- [a] Sébastien Bubeck. Convex optimization: Algorithms and complexity. Foundations and Trends R in
Machine Learning, 2015.

**Time Spent Reviewing:**

5

---

> ### Author Response · Authors · 2021-08-10
> **Thank you for your questions and feedback**
>
> Thank you for the detailed review and thoughtful feedback. Below we address specific questions and comments.
>
> **Q: Not clear mirror descent (MD) would be a special case of proximal connect: In my understanding, dual averaging is a special case of MD [a] and generalizing dual averaging does not mean MD would be a special case of the proposed method. Furthermore, MD supports nonlinear projections (such as tanh or softmax [1]) whereas the proposed formulation supports piecewise linear projections. So I'm not sure PC would be a generalization of the methods discussed in [1]. Please clarify.**
>
> A: We apologize for the confusion. We never claimed that MD (in its full generality) is a special case of PC, and Bubeck in his Section 4.4 [a] showed that DA is a lazy variant (rather than a special case) of MD. What we meant in Table 1 is that the proposed (univariate) MD method in Ref [1] (for NN quantization) is a special case of PC. Recall the MD update (using our notation) in Ref [1] (see their Eq (12) on page 5): $\\mathbf{w}_{t+1}^\\star = \\mathbf{w}_t^\\star - \\eta \\nabla f(\\mathrm{P}(\\mathbf{w}_t^\\star))$, where $\mathrm{P} = (\nabla \Phi)^{-1}$ for some mirror map $\Phi$ (see their proof of Theorem 1 on page 4).
>
> Since $\\Phi$ is taken to be strictly convex in [1], $\\mathrm{P}$ is monotone, single-valued and closed. Therefore, all conditions in our Theorem 3.1 are satisfied, and hence the quantizer $\\mathrm{P}$ in [1] (such as $\\tanh$) is a proximal map and their update above corresponds exactly to a specialization of PC (where $\\mathrm{P}$ is restricted to proximal quantizers given by the derivative of a smooth convex function). This is the power of our Theorem 3.1: although we may not immediately realize what the function $\\mathsf{r}$ looks like such that its proximal map $\\mathrm{P}_{\\mathsf{r}}$ is $\\tanh$, we know it exists which is all that matters.
>
> The above reasoning hinges on $\\mathrm{P}$ and $\\Phi$ being univariate. More generally, if a multivariate mirror map $\\Phi$ is 1-strongly convex (as is typical in MD), then it follows from a celebrated result of Moreau that $\\mathrm{P} := (\\nabla \\Phi)^{-1}$, being a nonexpansion, is again a proximal map (of some convex function). We do not need this result in our work but just in case one wonders what happens for multivariate $\\Phi$.
>
> We emphasize that all of our results hold for *any* proximal quantizer, whereas in contrast the quantizer in [1] is taken as the derivative of a smooth convex function, which is a strict subclass and in particular it does not allow for jumps. We designed a piece-wise $\\mathrm{P}$ in our experiments for simplicity and ease of comparison to prior work. We could have equally used $\\tanh$ or any other nonlinear proximal quantizer: all of our theoretical results still apply. This is also why we call PC a family of algorithms. We wish to point out that the theoretical result in [1, Theorem 2] only works for a bounded $\\nu$ (their $\\beta_k$) and convex losses, while we can fully justify the common practice of employing a diverging $\\nu$ (including the experiments in [1]) and accommodate nonconvex losses.
>
> We will add the above clarification to our final revision. Thank you for clearing this up.
>
> **Q: The main idea of this paper is in Eq. 8 (and Eq. 31, if I understand correctly) where the regularizer  is parametrized as opposed to ProxQuant [5]. May be this interpretation should be mentioned early so as to simplify the understanding of the final PC algorithm.**
>
> A: Thank you for this suggestion. We will incorporate the suggested adjustment and explain our proposed ProxConnect algorithm earlier in the paper.

---

> ### Author Response · Authors · 2021-08-31
> **Post-rebuttal update comment**
>
> We thank the reviewer for their active participation in the review process and we are more than happy to include the requested discussion in the revised version of our paper.

---

### Decision · Program_Chairs · 2021-09-27

**Decision:**

Accept (Poster)

**Comment:**

After the discussion, all reviewers recommend accept (7). For example, one reviewers emphasized "the submission made a non-trivial contribution upon existing works (Li et al., Bai et al.), in the sense that it formally shows the convergence for BinaryConnect for non-convex problems.". I find this alone sufficient novel. Also, the unification and extension of previously suggested algorithms are quite interesting, and the empirical results are not bad for a theoretical paper. The only concern raising from the reviews seems to be that the paper is dense, and I hope the authors can improve improve readability, if possible.